# Active Negative Loss Functions for Learning with Noisy Labels

**Xichen Ye**
Shanghai University
Shanghai, China
yexichen0930@shu.edu.cn

**Xiaoqiang Li** *
Shanghai University
Shanghai, China
xqli@shu.edu.cn

**Songmin Dai**
Shanghai University
Shanghai, China
laodar@shu.edu.cn

**Tong Liu**
Shanghai University
Shanghai, China
tong_liu@shu.edu.cn

**Yan Sun**
Shanghai University
Shanghai, China
yansun@shu.edu.cn

**Weiqin Tong**
Shanghai University
Shanghai, China
wqtong@shu.edu.cn

## Abstract

Robust loss functions are essential for training deep neural networks in the presence of noisy labels. Some robust loss functions use Mean Absolute Error (MAE) as its necessary component. For example, the recently proposed Active Passive Loss (APL) uses MAE as its passive loss function. However, MAE treats every sample equally, slows down the convergence and can make training difficult. In this work, we propose a new class of theoretically robust passive loss functions different from MAE, namely *Normalized Negative Loss Functions* (NNLFs), which focus more on memorized clean samples. By replacing the MAE in APL with our proposed NNLFs, we improve APL and propose a new framework called *Active Negative Loss* (ANL). Experimental results on benchmark and real-world datasets demonstrate that the new set of loss functions created by our ANL framework can outperform state-of-the-art methods. The code is available at https://github.com/Virusdoll/Active-Negative-Loss.

## 1 Introduction

Relying on large-scale datasets with high quality annotations, such as ImageNet [1], deep neural networks (DNNs) achieve good performance in various supervised classification tasks. However, in practice, the process of labeling large-scale datasets is costly and inevitably introduces noisy (mislabeled) samples. Moreover, empirical studies show that over-parameterized DNNs can easily fit a randomly labeled dataset [2], which implies that DNNs may have a poor evaluation performance when trained on a noisy dataset. As a result, noisy label learning has received a lot of attention.

Different approaches have been proposed to solve the noisy label learning problem, and one popular research line is to design noise-robust loss functions, which is also the main focus of this paper. Ghosh et al. [3] have theoretically proved that, symmetric loss functions such as Mean Absolute Error (MAE), are robust to noise, while others like commonly used Cross Entropy (CE) are not. However, MAE treats every sample equally, leading to significantly longer training time before convergence and even making learning difficult, which suggests that MAE is not suitable for training DNNs with challenging datasets [4]. Motivated by this, several works proposed partially robust loss functions, including Generalized Cross Entropy (GCE) [4], a generalized mixture of CE and MAE, and Symmetric Cross Entropy (SCE) [5], a combination of CE and a scaled MAE, Reverse Cross

---

*Corresponding Author

37th Conference on Neural Information Processing Systems (NeurIPS 2023).

Entropy (RCE). Recently, Active Passive Loss (APL) [6] framework has been proposed to create fully robust loss functions.

APL is one of the state-of-the-art methods, and shows that any loss function can be made robust to noisy labels by a simple normalization operation. Moreover, to address the underfitting problem of normalized loss functions, APL first characterizes existing loss functions into two types and then combines them. The two types are 1) "Active" loss, which only explicitly maximizes the probability of being in the labeled class, and 2) "Passive" loss, which also explicitly minimizes the probabilities of being in other classes. However, by investigating several robust loss functions created by the APL framework, we find that their passive loss functions are always scaled versions of MAE. As we mentioned before, MAE is not conducive to training. As a result, to address the underfitting problem, APL combines active loss functions with MAE, which again may lead to difficulty in training and further limits its performance.

The fact that APL still struggles with MAE motivates us to investigate new robust passive loss functions. In this paper, we propose a new class of passive loss functions different from MAE, called *Negative Loss Functions* (NLFs). We show that, by combining 1) complementary label learning [7, 8] and 2) a simple "vertical flipping" operation, any active loss function can be made into a passive loss function. Moreover, to make it theoretically robust to noisy labels, we further apply the normalization operation on NLFs to obtain *Normalized Negative Loss Functions* (NNLFs). By replacing the MAE in APL with NNLF, we propose a novel framework called *Active Negative Loss* (ANL). ANL combines a normalized active loss function and a NNLF to build a new set of noise-robust loss functions, which can be seen as an improvement of APL. We show that under our proposed ANL framework, several commonly-used loss functions can be made robust to noisy labels while ensuring sufficient fitting ability to achieve state-of-the-art performance for training DNNs with noisy datasets. Our key contributions are highlighted as follows:

- We provide a method to build a new class of robust passive loss functions called *Normalized Negative Loss Function*s (NNLFs). By replacing the MAE in APL with our proposed NNLFs, we propose a novel framework, *Active Negative Loss* (ANL), to construct a new set of robust loss functions.

- We demonstrate the theoretical robustness of our proposed NNLFs and ANL to noisy labels, and discuss how replacing the MAE in APL with our NNLFs enhances performance in noisy label learning.

- Our empirical results show that the new set of loss functions, created using our proposed ANL framework, outperform existing state-of-the-art methods.

## 2 Preliminaries

### 2.1 Risk Minimization and Label Noise Model

Consider a typical K-class classification problem. Let $\mathcal{X} \subset \mathbb{R}^d$ be the $d$-dimensional feature space from which the samples are drawn, and $\mathcal{Y} = [k] = \{1, \cdots, K\}$ be the label space. Given a clean training dataset, $\mathcal{S} = \{(\boldsymbol{x}_n, y_n)\}_{n=1}^N$, where each $(\boldsymbol{x}_n, y_n)$ is drawn *i.i.d.* from an unknown distribution, $\mathcal{D}$, over $\mathcal{X} \times \mathcal{Y}$. We denote the distribution over different labels for sample $\boldsymbol{x}$ by $\boldsymbol{q}(k|\boldsymbol{x})$, and $\sum_{k=1}^K \boldsymbol{q}(k|\boldsymbol{x}) = 1$. Since there is only one corresponding label $y$ for a $\boldsymbol{x}$, we have $\boldsymbol{q}(y|\boldsymbol{x}) = 1$ and $\boldsymbol{q}(k \neq y|\boldsymbol{x}) = 0$.

A classifier, $h(\boldsymbol{x}) = \arg\max_i f(\boldsymbol{x})_i$, where $f : \mathcal{X} \to \mathcal{C}, \mathcal{C} \subseteq [0,1]^K, \forall \boldsymbol{c} \in \mathcal{C}, \mathbf{1}^T \boldsymbol{c} = 1$, is a function that maps feature space to label space. In this work, we consider $f$ as a DNN ending with a softmax output layer. For each sample $\boldsymbol{x}$, $f(\boldsymbol{x})$ computes its probability $\boldsymbol{p}(k|\boldsymbol{x})$ of each label $k \in \{1, \cdots, K\}$, and $\sum_{k=1}^K \boldsymbol{p}(k|\boldsymbol{x}) = 1$. Throughout this paper, as a notation, we call $f$ itself the classifier. Training a classifier $f$ is to find an optimal classifier $f^*$ that minimize the empirical risk defined by a loss function: $\sum_{n=1}^N \mathcal{L}(f(\boldsymbol{x}_n), y_n)$, where $\mathcal{L} : \mathcal{C} \times \mathcal{Y} \to \mathbb{R}^+$ is a loss function, and $\mathcal{L}(f(\boldsymbol{x}), k)$ is the loss of $f(\boldsymbol{x})$ with respect to label $k$.

When label noise is present, our model can only access a corrupted dataset $\mathcal{S}_\eta = \{(\boldsymbol{x}_n, \hat{y}_n)\}_{n=1}^N$, where each sample is drawn *i.i.d.* from an unknown distribution, $\mathcal{D}_\eta$. In this paper, we consider a popular approach for modeling label noise, which simply assumes that, given the true label $y$, the

corruption process is conditionally independent of input features $\boldsymbol{x}$ [9]. So we can formulate noisy label $\hat{y}$ as:

$$\hat{y} = \begin{cases} y & \text{with probability } (1 - \eta_y) \\ j, j \in [k], j \neq y & \text{with probability } \eta_{yj} \end{cases}, \tag{1}$$

where $\eta_{yj}$ denotes the probability that true label $y$ is corrupted into label $j$, and $\eta_y = \sum_{j \neq y} \eta_{yj}$ denotes the noise rate of label $y$. Under our assumption of label noise model, label noise can be either *symmetric* or *asymmetric*. The noise is called *symmetric*, if $\eta_{ij} = \frac{\eta_i}{K-1}, \forall j \neq y$ and $\eta_i = \eta, \forall i \in [k]$, where $\eta$ is a constant. And for *asymmetric* noise, $\eta_{ij}$ is conditioned on both the true label $i$ and corrupted label $j$.

## 2.2 Active Passive Loss Functions

Ghosh et al. [3] have shown, under some mild assumptions, a loss function $\mathcal{L}$ is noise tolerant if it is symmetric: $\sum_{k=1}^{K} \mathcal{L}(f(\boldsymbol{x}), k) = C, \forall \boldsymbol{x} \in \mathcal{X}$, where $C$ is some constant. Based on this, Ma et al. [6] proposed the normalized loss functions, which normalize a loss function $\mathcal{L}$ by:

$$\mathcal{L}_{\text{norm}} = \frac{\mathcal{L}(f(\boldsymbol{x}), y)}{\sum_{k=1}^{K} \mathcal{L}(f(\boldsymbol{x}), k)}. \tag{2}$$

This simple normalization operation can make any loss function robust to noisy labels, since we always have $\sum_{k}^{K} \mathcal{L}_{\text{norm}} = 1$. For example, the Normalized Cross Entropy (NCE) is:

$$NCE = \frac{\sum_{k=1}^{K} \boldsymbol{q}(k|\boldsymbol{x})(-\log \boldsymbol{p}(k|\boldsymbol{x}))}{\sum_{j=1}^{K} \sum_{k=1}^{K} \boldsymbol{q}(y = j|\boldsymbol{x}) \log \boldsymbol{p}(k|\boldsymbol{x})}. \tag{3}$$

Similarly, we can normalize FL, MAE, and RCE to obtain Normalized Focal Loss (NFL), Normalized Mean Absolute Error (NMAE), and Normalized Reverse Cross Entropy (NRCE), respectively.

But a normalized loss function alone suffers from the underfitting problem. To address this, Ma et al. [6] characterize existing loss functions into two types: *Active* and *Passive*. Denote the function of loss $\mathcal{L}(f(\boldsymbol{x}), y)$ by $\ell(f(\boldsymbol{x}), k)$, that is $\mathcal{L}(f(\boldsymbol{x}), y) = \sum_{k=1}^{K} \ell(f(\boldsymbol{x}), k)$ (e.g., let $\mathcal{L}$ be CE, then $\mathcal{L}(f(\boldsymbol{x}), y) = \sum_{k=1}^{K} \boldsymbol{q}(k|\boldsymbol{x})(-\log \boldsymbol{p}(k|\boldsymbol{x}))$, and $\ell(f(\boldsymbol{x}), k) = \boldsymbol{q}(k|\boldsymbol{x})(-\log \boldsymbol{p}(k|\boldsymbol{x})))$, we have the following definitions:

**Definition 1** (Active loss function). $\mathcal{L}_{\text{Active}}$ is an active loss function if $\forall (\boldsymbol{x}, y) \in \mathcal{D}, \forall k \neq y, \ell(f(\boldsymbol{x}), k) = 0$.

**Definition 2** (Passive loss function). $\mathcal{L}_{\text{Passive}}$ is a passive loss function if $\forall (\boldsymbol{x}, y) \in \mathcal{D}, \exists k \neq y, \ell(f(\boldsymbol{x}), k) \neq 0$.

Active loss functions only explicitly maximize $\boldsymbol{p}(y|\boldsymbol{x})$, the classifier's output probability at the class position specified by the label $y$. In contrast, passive loss functions also explicitly minimize $\boldsymbol{p}(k \neq y|\boldsymbol{x})$, the probability at least one other class positions. Accordingly, the active loss functions include CE, FL, NCE, and NFL, while the passive loss functions include MAE, RCE, NMAE, and NRCE. These two types of loss functions can mutually boost each other to mitigate underfitting, and we refer the reader to [6] for more detailed discussions. By combining them, Ma et al. proposed the Active Passive Loss (APL):

$$\mathcal{L}_{\text{APL}} = \alpha \cdot \mathcal{L}_{\text{Active}} + \beta \cdot \mathcal{L}_{\text{Passive}}, \tag{4}$$

where $\alpha, \beta > 0$ are parameters. As an example, by combining NCE and RCE, Ma et al get NCE+RCE, one of the state-of-the-art methods.

## 2.3 APL struggles with MAE

As shown in the previous subsection, there are four passive loss functions available to APL, including MAE, RCE, NMAE, and NRCE. However, we can show that all these passive loss functions are scaled versions of MAE. Specifically, $NMAE = \frac{1}{2(K-1)} \cdot MAE$, $RCE = -\frac{A}{2} \cdot MAE$, and $NRCE = \frac{1}{2(K-1)} \cdot MAE$ (detailed derivations can be found in appendix A.1). Thus, we can rewrite APL as follows:

$$\mathcal{L}_{\text{APL}} = \alpha \cdot \mathcal{L}_{\text{Active}} + \beta \cdot MAE. \tag{5}$$

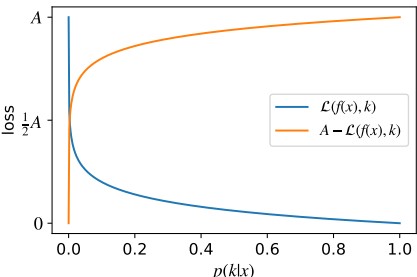

Figure 1: "Vertical flipping" operation. $\mathcal{L}(f(\boldsymbol{x}), k)$ is an active loss function, e.g., CE. Loss function $A - \mathcal{L}(f(\boldsymbol{x}), k)$ is obtained by flipping $\mathcal{L}(f(\boldsymbol{x}), k)$ vertically with axis loss $= \frac{1}{2}A$.

This indicates that MAE is a necessary component of the current APL. However, as we mentioned before, MAE requires longer training time and even makes learning difficult. Thus, on the one hand, APL needs passive loss functions to mitigate active loss functions underfitting, yet on the other hand, MAE is not training friendly, which may limit the performance of APL. This motivates us to investigate new robust passive loss functions.

## 3 Active Negative Loss Functions

### 3.1 Method

**Normalized Negative Loss Functions.** Our goal is to find a method that creates robust passive loss functions from existing active loss functions. This method must consist of three components that: 1) let the loss function optimize the classifier's output probability for at least one other class position that is not specified by the label $y$, 2) let the loss function minimize the classifier's output probability instead of maximizing it, and 3) let the loss function robust to noisy labels. Inspired by NLNL [10] and APL [6], we use 1) complementary label learning, 2) "vertical flipping" and 3) normalization operation as these three components respectively.

Complementary label learning [7, 8] is an indirect learning method for training CNNs, which randomly selects complementary labels and trains the CNN to recognize that "input does not belong to this complementary label". Unlike the usual training approach, complementary label learning focuses on the loss of classifier's predictions with complementary labels, which naturally fits with the passive loss function. Here, we only use its basic idea of letting the loss function focus on all classes $\{1, \cdots, K\}$ except the labeled class $y$.

"Vertical flipping" is a simple operation that can convert the loss function from "maximizing" to "minimizing". As shown in the fig. 1, given an active loss function $\mathcal{L}(f(\boldsymbol{x}), k)$, the new loss function $A - \mathcal{L}(f(\boldsymbol{x}), k)$ is obtained by flipping $\mathcal{L}(f(\boldsymbol{x}), k)$ vertically with axis loss $= \frac{1}{2}A$. It should be noted that, $A - \mathcal{L}(f(\boldsymbol{x}), k)$ is the opposite of $\mathcal{L}(f(\boldsymbol{x}), k)$, and it focuses on optimizing $\boldsymbol{p}(k|\boldsymbol{x})$ to 0.

Based on these two components, given an active loss function $\mathcal{L}$, we propose Negative Loss Functions (NLFs) as follows:

$$\mathcal{L}_{\text{neg}}(f(\boldsymbol{x}), y) = \sum_{k=1}^{K}(1 - \boldsymbol{q}(k|\boldsymbol{x}))(A - \mathcal{L}(f(\boldsymbol{x}), k)), \tag{6}$$

$$A = \mathcal{L}([\cdots, \boldsymbol{p}_{\min}, \cdots]^T, y). \tag{7}$$

Here, $[\cdots, \boldsymbol{p}_{\min}, \cdots]^T$ is some probability distribution that may be output by the classifier $f$, where $\boldsymbol{p}(y|\boldsymbol{x}) = \boldsymbol{p}_{\min}$, the minimum value of $\boldsymbol{p}(k|\boldsymbol{x})$ (e.g., 0). Therefore $A$ is some constant, the maximum loss value of $\mathcal{L}$. In practice, setting $\boldsymbol{p}_{\min} = 0$ could cause some computational problems, for example, if $\mathcal{L}$ is CE and $\boldsymbol{p}_{\min} = 0$, then $A = -\log 0 = +\infty$. So in this paper, unless otherwise specified, we define $\boldsymbol{p}_{\min} = 1 \times 10^{-7}$. This technique is similar to the clipping operation implemented by most deep learning frameworks.

Our proposed NLF can transform any active loss function into a passive loss function, where 1) $(1 - \boldsymbol{q}(k|\boldsymbol{x}))$ ensures that the loss function focuses on classes $\{1, \cdots, K\} \setminus \{y\}$, and 2) $(A - \mathcal{L}(f(\boldsymbol{x}), k))$ ensures that the loss function aims to minimize the output probability $\boldsymbol{p}(k|\boldsymbol{x})$.

Next, to make our proposed passive loss functions robust to noisy labels, we perform a normalization operation on NLFs. Given an active loss function $\mathcal{L}$, we propose Normalized Negative Loss Functions (NNLFs) as follows:

$$\mathcal{L}_{\text{nn}}(f(\boldsymbol{x}), y) = 1 - \frac{A - \mathcal{L}(f(\boldsymbol{x}), y)}{\sum_{k=1}^{K} A - \mathcal{L}(f(\boldsymbol{x}), k)}, \tag{8}$$

where $A$ has the same definition as eq. (7). The detailed derivation of NNLFs can be found in appendix A.2. Additionally, NNLFs have the property that $\mathcal{L}_{\text{nn}} \in [0, 1]$. Accordingly, we can create NNLFs from active loss functions as follows.

The Normalized Negative Cross Entropy (NNCE) is:

$$NNCE = 1 - \frac{A - (-\log \boldsymbol{p}(y|\boldsymbol{x}))}{\sum_{k=1}^{K} A - (-\log \boldsymbol{p}(k|\boldsymbol{x}))}, \tag{9}$$

where $A = -\log \boldsymbol{p}_{\min}$.

The Normalized Negative Focal Loss (NNFL) is:

$$NNFL = 1 - \frac{A - (-(1 - \boldsymbol{p}(y|\boldsymbol{x}))^{\gamma} \log \boldsymbol{p}(y|\boldsymbol{x}))}{\sum_{k=1}^{K} A - (-(1 - \boldsymbol{p}(k|\boldsymbol{x}))^{\gamma} \log \boldsymbol{p}(k|\boldsymbol{x}))}, \tag{10}$$

where $A = -(1 - \boldsymbol{p}_{\min})^{\gamma} \log \boldsymbol{p}_{\min}$.

**ANL Framework.** We can now create new robust loss functions by replacing the MAE in APL with our proposed NNLF. Given an active loss function $\mathcal{L}$, we propose the Active Negative Loss (ANL) functions as follows:

$$\mathcal{L}_{\text{ANL}} = \alpha \cdot \mathcal{L}_{\text{norm}} + \beta \cdot \mathcal{L}_{\text{nn}}. \tag{11}$$

Here, $\alpha$ and $\beta$ are parameters greater than 0, $\mathcal{L}_{\text{norm}}$ denotes the normalized $\mathcal{L}$ and $\mathcal{L}_{\text{nn}}$ denotes the Normalized Negative Loss Function corresponding to $\mathcal{L}$. Accordingly, we can create ANL from the two mentioned active loss functions as follows.

For Cross Entropy (CE), we have ANL-CE:

$$\mathcal{L}_{\text{ANL-CE}} = \alpha \cdot \mathcal{L}_{\text{NCE}} + \beta \cdot \mathcal{L}_{\text{NNCE}}. \tag{12}$$

For Focal Loss (FL), we have ANL-FL:

$$\mathcal{L}_{\text{ANL-FL}} = \alpha \cdot \mathcal{L}_{\text{NFL}} + \beta \cdot \mathcal{L}_{\text{NNFL}}. \tag{13}$$

### 3.2 Robustness to noisy labels

**NNLFs are symmetric.** We first prove that our proposed Normalized Negative Loss Functions (NNLFs) are symmetric. Detailed proofs can be found in appendix B.

**Theorem 1.** *Normalized negative loss function $\mathcal{L}_{nn}$ is symmetric.*

**NNLFs are robust to noise.** In reference to Theorem 1 and Theorem 3 from [3], it has been proven that symmetric loss functions, under some mild assumptions, exhibit noise tolerant in the face of both symmetric and asymmetric noise. Given that our NNLFs fall under the category of symmetric loss functions, they inherently possess the attribute of noise tolerant.

**ANL is robust to noise.** In light of Lemma 3 from [6], it is understood that the combination of two noise tolerant loss functions retains the noise tolerant attribute. It is noteworthy that both $\mathcal{L}_{\text{norm}}$ and $\mathcal{L}_{\text{nn}}$ within our ANL are noise tolerant, which makes ANL as a whole noise tolerant.

### 3.3 NNLFs focus more on well-learned samples

As shown in the fig. 2, by replacing MAE with our proposed NNCE, NCE+NNCE and ANL-CE show better fitting ability. This raises the question: *why does NNLF perform better than MAE?* In the following, taking NNCE as an example, we analysis the gradients of MAE and NNCE to provide a preliminary answer to this question. Detailed derivations and proofs can be found in appendix C.

The gradient of the MAE with respect to the classifier's output probability can be derived as:

$$\frac{\partial \mathcal{L}_{\text{MAE}}}{\partial \boldsymbol{p}(j|\boldsymbol{x})} = \begin{cases} 1, & j \neq y \\ -1, & j = y. \end{cases} \tag{14}$$

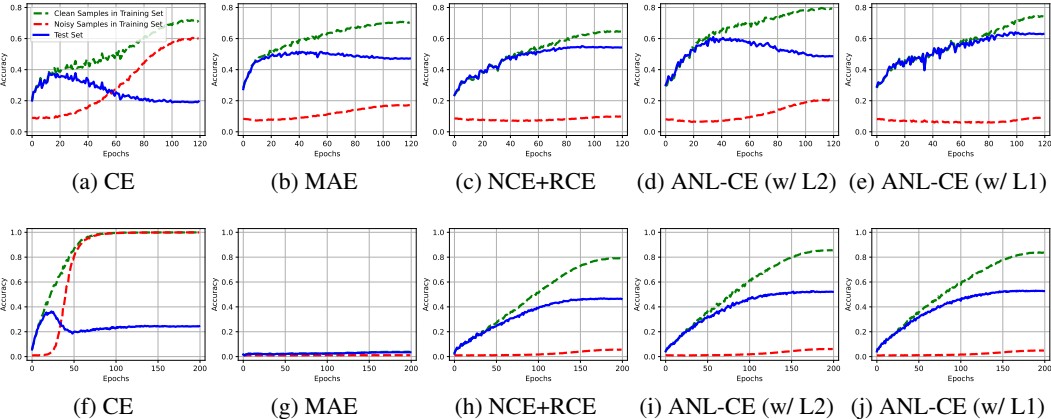

Figure 2: Training and test accuracies of different loss functions. (a) - (e): CIFAR-10 under 0.8 symmetric noise. (f) - (j): CIFAR-100 under 0.6 symmetric noise. The accuracies of noisy samples in training set should be as low as possible, since they are mislabeled.

The gradient of the NNCE with respect to the classifier's output probability can be derived as:

$$
\frac{\partial \mathcal{L}_{\text{NNCE}}}{\partial \boldsymbol{p}(j|\boldsymbol{x})} = \begin{cases} \frac{1}{\boldsymbol{p}(j|\boldsymbol{x})} \cdot \frac{A + \log \boldsymbol{p}(y|\boldsymbol{x})}{\left( \sum_{k=1}^{K} A + \log \boldsymbol{p}(k|\boldsymbol{x}) \right)^2}, & j \neq y \\ -\frac{1}{\boldsymbol{p}(y|\boldsymbol{x})} \cdot \frac{\sum_{k \neq y} A + \log \boldsymbol{p}(k|\boldsymbol{x})}{\left( \sum_{k=1}^{K} A + \log \boldsymbol{p}(k|\boldsymbol{x}) \right)^2}, & j = y. \end{cases} \tag{15}
$$

For the purpose of analysis, we consider how the gradients of NNCE would differ from MAE in the following two scenarios: 1) given the classifier's output probability of sample $\boldsymbol{x}$, we analyze the difference in gradient for each class, 2) given the classifier's output probabilities of sample $\boldsymbol{x}_1$ and $\boldsymbol{x}_2$, we analyze the difference in gradient between these two samples.

**Theorem 2.** *Given the classifier's output probability $\boldsymbol{p}(\cdot|\boldsymbol{x})$ for sample $\boldsymbol{x}$ and normalized negative cross entropy $\mathcal{L}_{\text{NNCE}}$. If $\boldsymbol{p}(j_1|\boldsymbol{x}) < \boldsymbol{p}(j_2|\boldsymbol{x})$, $j_1 \neq j_2 \neq y$, then $\frac{\partial \mathcal{L}_{\text{NNCE}}}{\partial \boldsymbol{p}(j_1|\boldsymbol{x})} > \frac{\partial \mathcal{L}_{\text{NNCE}}}{\partial \boldsymbol{p}(j_2|\boldsymbol{x})}$.*

**Theorem 3.** *Given the classifier's output probabilities $\boldsymbol{p}(\cdot|\boldsymbol{x}_1)$ and $\boldsymbol{p}(\cdot|\boldsymbol{x}_2)$ of sample $\boldsymbol{x}_1$ and $\boldsymbol{x}_2$, where $\boldsymbol{p}(y|\boldsymbol{x}_1) \geq \boldsymbol{p}(k|\boldsymbol{x}_1)$, $\boldsymbol{p}(y|\boldsymbol{x}_2) \geq \boldsymbol{p}(k|\boldsymbol{x}_2)$, $\forall k \in \{1, \cdots, K\}$, $\boldsymbol{p}(j|\boldsymbol{x}_1) = \boldsymbol{p}(j|\boldsymbol{x}_2)$, $j \neq y$, and normalized negative cross entropy $\mathcal{L}_{\text{NNCE}}$. If $\boldsymbol{p}(y|\boldsymbol{x}_1) > \boldsymbol{p}(y|\boldsymbol{x}_2)$ and $\boldsymbol{p}(k|\boldsymbol{x}_1) \leq \boldsymbol{p}(k|\boldsymbol{x}_2)$, $\forall k \in \{1, \cdots, K\} \setminus \{j, y\}$, then $\frac{\partial \mathcal{L}_{\text{NNCE}}}{\partial \boldsymbol{p}(j|\boldsymbol{x}_1)} > \frac{\partial \mathcal{L}_{\text{NNCE}}}{\partial \boldsymbol{p}(j|\boldsymbol{x}_2)}$.*

theorem 2 and theorem 3 demonstrate that, for the gradient of the non-labeled classes, our NNCE focuses more on the classes and samples that have been well learned compared to MAE, which treats every class and sample equally This property may enhance the model's performance in noisy label learning. Some studies [11] have shown that during the training process, DNNs would first memorize clean samples and then noisy samples. According to the property revealed by theorem 2 and theorem 3, apart from robustness, our NNLF may potentially help the model to continuously learn the clean samples that the model has memorized in the previous stages of training and ignore the unmemorized noisy samples.

## 4 Experiments

In this section, we empirically investigate our proposed ANL functions on benchmark datasets, including MNIST [12], CIFAR-10/CIFAR-100 [13] and a real-world noisy dataset WebVision [14].

### 4.1 Empirical Understandings

In this subsection, we explore some properties of our proposed loss functions. Unless otherwise specified, all detailed experimental settings are the same as those in section 4.2. More experiments and discussions about gradient, parameter analysis, and $\boldsymbol{p}_{\min}$ can be found in the appendix D.1.

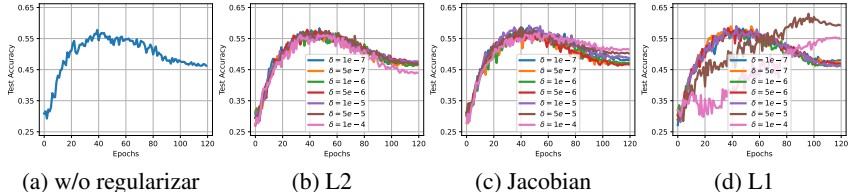

| (a) w/o regularizar | (b) L2 | (c) Jacobian | (d) L1 |

Figure 3: Test accuracies of ANL-CE on CIFAR-10 under 0.8 symmetric noise with different regularization methods and different parameters. $\delta$ is the weight of regularization term of ANL-CE.

Table 1: Test accuracies (%) of different methods on CIFAR-10 datasets with clean and symmetric ($\eta \in \{0.2, 0.4, 0.6, 0.8\}$) label noise. The top-1 best results are in **bold**.

| Methods | Clean ($\eta = 0.0$) | $\eta = 0.2$ | $\eta = 0.4$ | $\eta = 0.6$ | $\eta = 0.8$ |
|---------|---------------------|--------------|--------------|--------------|--------------|
| NCE     | 88.68               | 81.65        | 74.80        | 63.14        | 37.52        |
| NNCE    | 91.51               | 90.09        | 86.91        | 82.16        | 57.06        |
| ANL-CE  | 91.66±0.04          | **90.02±0.23** | **87.28±0.02** | **81.12±0.30** | **61.27±0.55** |

**Overfitting problem.** In practice, we find that ANL can lead to overfitting in some experimental settings. To motivate this problem, as an example, we train networks using different loss functions on CIFAR-10 under 0.8 symmetric noise and CIFAR-100 under 0.6 symmetric noise, and the experimental results are shown in fig. 2. As can be observed, in the setting of CIFAR-10 under 0.8 symmetric noise, the training set accuracy of ANL-CE (w/ L2) keeps increasing while the test set accuracy keeps decreasing. We identify this problem as an *overfitting problem*.

It is worth noting that although overfitting occurs, unlike CE, the gap between the clean sample accuracies and the noisy sample accuracies of the training set does not shrink, which indicates that our ANL has some robustness to noisy labels even in the case of overfitting. Moreover, we conjecture that the overfitting is caused by the property of NNLF focusing more on well-learned samples. When the noise rate is high, one might assume that the model has been trained on only a fairly small number of clean samples, with a very large gradient, which can lead to overfitting.

**The choice of regularization method.** We also find that the commonly used L2 regularization may struggle to mitigate the overfitting problem. To address this, we decide to try using other regularization methods. We consider 2 other regularization methods: L1 and Jacobian [15, 16]. To compare the performance of these methods, we apply them to ANL-CE for training on CIFAR-10 under 0.8 symmetric noise. For simplicity and fair comparison, we use $\delta$ as the coefficient of the regularization term and consider it as an additional parameter of ANL. We keep $\alpha$ and $\beta$ the same as in section 4.2, tune $\delta$ for all three methods by following the parameter tuning setting in appendix D.2. We also train networks without using any regularization method. The results reported in fig. 3. As can be observed, among the three regularization methods, only L1 can somewhat better mitigate the overfitting problem. If not otherwise specified, all ANLs in this paper use L1 regularization.

**Robustness and fitting ability.** We conduct a set of experiments on CIFAR-10/-100 to verify the robustness and fitting ability of our proposed loss functions. We set the noise type to be symmetric and the noise rate to 0.8 for CIFAR-10 and 0.6 for CIFAR-100. In each setting, we train the network using different loss functions, including: 1) CE, 2) MAE, 3) NCE+RCE, 4) ANL-CE (w/ L2), and 5) ANL-CE (w/ L1). For ANL-CE (w/ L2), we set its parameters $\alpha$ and $\beta$ to be the same as ANL-CE (w/ L1) and set its weight decay to be the same as NCE+RCE.

As can be observed in fig. 2: 1) CE is not robust to noise, the accuracies of clean and noisy samples in the training set are continuously close to each other, 2) MAE is robust to noise, the accuracies of clean and noisy samples in the training set keep moving away from each other, but its fitting ability is insufficient, especially when the dataset becomes complex, 3) NCE+RCE is robust to noise and has better fitting ability compared to MAE, 4) ANL-CE (w/ L2) is robust to noise and has stronger fitting ability, but suffers from over-fitting. and 5) ANL-CE is robust to noise and mitigates the impact of overfitting to achieve the best performance. To summarize, our proposed loss functions are robust to noise, NNLF shows better fitting ability than MAE, and L1 regularization addresses the overfitting problem of NNLF.

Table 2: Test accuracies (%) of different methods on benchmark datasets with clean, symmetric ($\eta \in \{0.4, 0.6, 0.8\}$) or asymmetric ($\eta \in \{0.2, 0.3, 0.4\}$) label noise. The results (mean±std) are reported over 3 random runs under different random seeds (1, 2, 3). The top-2 best results are in **bold**.

| Datasets | Methods | Clean ($\eta$=0.0) | Symmetric Noise Rate ($\eta$) | | | Asymmetric Noise Rate ($\eta$) | | |
| | | | 0.4 | 0.6 | 0.8 | 0.2 | 0.3 | 0.4 |
|---|---|---|---|---|---|---|---|---|
| MNIST | CE | 99.20±0.02 | 74.46±0.28 | 49.19±0.05 | 22.51±0.23 | 94.02±0.18 | 88.90±0.07 | 81.79±0.34 |
| | MAE | 99.16±0.03 | 98.80±0.02 | 97.69±0.20 | 70.35±1.16 | **99.11±0.03** | 98.42±0.09 | 87.40±4.01 |
| | GCE [4] | 99.18±0.01 | 96.81±0.13 | 80.86±0.31 | 33.59±0.48 | 96.59±0.07 | 88.99±0.27 | 81.91±0.58 |
| | SCE [5] | 99.30±0.07 | 97.48±0.16 | 88.35±0.77 | 48.28±0.81 | 97.95±0.23 | 94.00±0.41 | 84.54±0.14 |
| | NLNL [10] | 98.61±0.13 | 97.17±0.09 | 95.42±0.30 | 86.34±1.43 | 98.35±0.01 | 97.51±0.15 | 95.84±0.26 |
| | NCE+RCE [6] | 99.43±0.02 | 98.53±0.09 | 95.61±0.12 | 74.04±1.83 | 98.79±0.10 | 95.16±0.08 | 91.36±0.22 |
| | NCE+AGCE [17] | 99.10±0.03 | **98.91±0.04** | **98.50±0.07** | **96.93±0.13** | 99.04±0.02 | **98.94±0.03** | **98.41±0.04** |
| | **ANL-CE** | 99.08±0.05 | 98.84±0.05 | 98.42±0.08 | **96.62±0.12** | 99.04±0.04 | 98.91±0.07 | **98.01±0.10** |
| | **ANL-FL** | 99.13±0.05 | **98.90±0.05** | **98.46±0.12** | 95.73±0.22 | **99.05±0.09** | **98.93±0.02** | 98.18±0.01 |
| CIFAR-10 | CE | 90.38±0.11 | 58.19±0.21 | 38.75±0.19 | 19.09±0.35 | 83.00±0.33 | 78.15±0.17 | 73.69±0.20 |
| | MAE | 89.15±0.27 | 81.76±3.17 | 76.82±0.84 | 46.42±3.66 | 79.63±0.74 | 67.35±3.41 | 57.36±2.37 |
| | GCE [4] | 89.66±0.20 | 82.44±0.26 | 68.62±0.35 | 25.45±0.51 | 85.55±0.24 | 79.32±0.52 | 72.83±0.17 |
| | SCE [5] | 91.38±0.12 | 79.96±0.25 | 62.16±0.33 | 27.98±0.98 | 86.22±0.44 | 80.20±0.20 | 74.01±0.52 |
| | NLNL [10] | 90.73±0.20 | 63.90±0.44 | 50.68±0.47 | 29.53±1.55 | 84.74±0.08 | 81.26±0.43 | 76.97±0.52 |
| | NCE+RCE [6] | 90.94±0.01 | 86.03±0.13 | 79.89±0.11 | 55.52±2.74 | 88.36±0.13 | 84.84±0.16 | **77.75±0.37** |
| | NCE+AGCE [17] | 91.08±0.06 | 86.16±0.10 | 80.14±0.27 | 55.62±4.78 | 88.48±0.09 | 84.79±0.15 | **78.60±0.41** |
| | **ANL-CE** | 91.66±0.04 | **87.28±0.02** | 81.12±0.30 | **61.27±0.55** | **89.13±0.11** | 85.52±0.24 | 77.63±0.31 |
| | **ANL-FL** | 91.79±0.19 | **87.25±0.11** | **81.67±0.19** | **61.22±0.85** | **89.09±0.31** | **85.81±0.23** | 77.73±0.31 |
| CIFAR-100 | CE | 71.14±0.38 | 40.72±0.74 | 22.98±0.07 | 7.55±0.21 | 58.25±1.00 | 50.30±0.19 | 41.53±0.34 |
| | MAE | 7.35±1.19 | 3.61±0.21 | 3.63±0.35 | 2.83±1.35 | 6.19±0.42 | 5.82±0.96 | 3.96±0.35 |
| | GCE [4] | 61.62±0.43 | 56.46±0.95 | 46.27±1.30 | 19.51±0.86 | 59.06±0.46 | 53.88±0.96 | 41.51±0.52 |
| | SCE [5] | 70.80±0.37 | 39.84±0.19 | 21.97±0.92 | 7.87±0.48 | 57.78±0.83 | 50.15±0.12 | 41.33±0.86 |
| | NLNL [10] | 68.72±0.60 | 30.29±1.64 | 16.60±0.90 | 11.01±2.48 | 50.19±0.56 | 42.81±1.13 | 35.10±0.20 |
| | NCE+RCE [6] | 68.22±0.28 | 57.97±0.30 | 46.26±1.07 | 25.65±0.51 | 62.77±0.53 | 55.62±0.56 | 42.46±0.42 |
| | NCE+AGCE [17] | 68.61±0.12 | 59.74±0.68 | 47.96±0.44 | 24.13±0.07 | 64.05±0.25 | 56.36±0.59 | 44.90±0.62 |
| | **ANL-CE** | 70.68±0.23 | **61.80±0.50** | **51.52±0.53** | **28.07±0.28** | **66.27±0.19** | **59.76±0.34** | **45.41±0.68** |
| | **ANL-FL** | 70.40±0.15 | **61.73±0.48** | **51.32±0.34** | **27.97±0.58** | **66.26±0.44** | **59.68±0.86** | **46.65±0.04** |

Table 3: Top-1 validation accuracies(%) on clean ILSVRC12 and WebVision validation set of ResNet-50 models trained on WebVision using different methods. The top-2 best results are in **bold**.

| Methods | CE | GCE [4] | SCE [5] | NCE+RCE [6] | NCE+AGCE [17] | **ANL-CE** | **ANL-FL** |
|---|---|---|---|---|---|---|---|
| ILSVRC12 Val | 58.64 | 56.56 | 62.60 | 62.40 | 60.76 | **65.00** | **65.56** |
| WebVision Val | 61.20 | 59.44 | **68.00** | 64.92 | 63.92 | 67.44 | **68.32** |

**Active and passive parts separately.** In table 1, we show the results of the active and passive parts separately. We separately train NCE and NNCE on CIFAR-10 with different symmetric noise rates while maintaining the same parameters as ANL-CE. Specifically, for $\alpha \cdot$ NCE, we set $\alpha$ to 5.0 and for $\beta \cdot$ NNCE, we set $\beta$ to 5.0, while $\delta$ is set to $5 \times 10^{-5}$ for both. As indicated in the results, the test set accuracies of NNCE are very close to those of ANL-CE, except in the case with a 0.8 noise rate. This suggests that NNCE performs well on its own at low noise rates. However, at very high noise rates, a combination of active losses is needed to achieve better performance.

## 4.2 Evaluation on Benchmark Datasets

**Baselines.** We consider several state-of-the-art methods: 1) Generalized Cross Entropy (GCE) [4]; 2) Symmetric Cross Entropy (SCE) [5]; 3) Negative Learning for Noisy Labels (NLNL) [10]; 4) Active Passive Loss (APL) [6], including NCE+MAE, NCE+RCE, and NFL+RCE; 5) Asymmetric Loss Functions (AFLs) [17], including NCE+AEL, NCE+AGCE, and NCE+AUL. For our proposed ANL, we consider two loss functions: 1) ANL-CE and 2) ANL-FL. Additionally, we train networks using Cross Entropy (CE), Focal Loss (FL) [18], and Mean Absolute Error (MAE).

**Experimental Details.** The full experimental results and the detailed settings of noise generation, networks, training and parameters can be found in the appendix D.2.

**Results.** The main experimental results under symmetric and asymmetric label noise are reported in table 2. For more experimental results, please see appendix D.2. As can be observed, our ANL-CE and ANL-FL show significant improvement for most label noise settings of CIFAR-10/-100, especially when the data become more complex and the noise rate becomes larger. For example, on CIFAR-10 under 0.8 symmetric noise, our ANL-CE outperform the state-of-the-art method (55.62% of NCE+AGCE) by more than 5.0%. Overall, the experimental results demonstrate that our ANL can show outstanding performance on different datasets, noise types, and noise rates, which validates the effectiveness of our proposed NNLFs and ANL.

### 4.3 Evaluation on Real-world Noisy Labels

Here, we evaluate our proposed ANL methods on large-scale real-world noisy dataset WebVision [14], which contains more than 2.4 million web images crawled from the internet by using queries generated from the 1,000 class labels of the ILSVRC 2012 [1] benchmark. Here, we follow the "Mini" setting in [19], and only take the first 50 classes of the Google resized image subset. We evaluate the trained networks on the same 50 classes of both the ILSVRC 2012 validation set and the WebVision validation set, and these can be considered as clean validation sets. We compare our ANL-CE and ANL-FL with GCE, SCE, NCE+RCE, and NCE+AGCE. The experimental details can be found in the appendix D.3. The results are reported in table 3. As can be observed, our proposed methods outperform the existing loss functions. This verifies that our proposed ANL framework can help the trained model against real-world label noise.

Moreover, in addition to WebVision, to further validate the effectiveness of our method on real-world noisy datasets, we also conduct a set of experiments on CIFAR-10N/-100N [20], Animal-10N [21], and Clothing-1M [22]. The experimental details and results can be found in the appendix D.5, which demonstrate the effectiveness of our method on different real-world noisy datasets.

## 5 Limitations

We believe that the main limitation of our approach lies in the choice of regularization method. Although we have experimentally verified that L1 is the most efficient among the three regularization methods (L1, L2, and Jacobian), we lack further theoretical analysis of it. Furthermore, although we only consider relatively simple regularization methods for the sake of fair comparison, other regularization methods, such as dropout [23] or regmixup [24], might be more effective in mitigating the overfitting problem caused by NNLF. And we believe that a better solution to overfitting can further improve the performance of our method.

## 6 Related Work

In recent years, some robust loss-based methods have been proposed for robust learning with noisy labels. Here, we briefly review the relevant approaches. Ghosh et al. [3] theoretically proved that symmetric loss functions, such as MAE, are robust to label noise. Zhang and Sabuncu [4] proposed Generalized Cross Entropy (GCE), a generalization of CE and MAE. Wang et al. [5] suggested a combination of CE and scaled MAE, and proposed Symmetric Cross Entropy (SCE). Menon et al. [25] proposed composite loss-based gradient clipping and applied it to CE to obtain PHuber-CE. Ma et al. [6] proposed Active Passive Loss (APL) to create fully robust loss functions. Feng et al. [26] applied the Taylor series to derive an alternative representation of CE and proposed Taylor-CE accordingly. Zhou et al. [17] proposed Asymmetric Loss Functions (ALFs) to overcome the symmetric condition. Inspired by complementary label learning, NLNL [10] and JNPL [27] use complementary labels to reduce the risk of providing the wrong information to the model.

## 7 Conclusion

In this paper, we propose a new class of theoretically robust passive loss functions different from MAE, which we refer to as *Normalized Negative Loss Functions* (NNLFs). By replacing the MAE in APL with our NNLF, we propose *Active Negative Loss* (ANL), a robust loss function framework with stronger fitting ability. We theoretically demonstrate that our NNLFs and ANLs are robust to noisy labels and also highlight the property that NNLFs focus more on well-learned samples. We found in our experiments that NNLFs have a potential overfitting problem, and we suggest using L1 regularization to mitigate it. Experimental results verified that our ANL can outperform the state-of-the-art methods on benchmark datasets.

## Acknowledgements

This work is supported in part by Shanghai Science and Technology Committee under grant No. 21511100600 and No. 22511106005.

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

## A  Loss functions

### A.1  NMAE, RCE, and NRCE are scaled versions of MAE

Here, inspired by [5, 6], we show how to reformulate NMAE, RCE, and NRCE as scaled versions of MAE. First, for convenience, we reformulate the Mean Absolute Error (MAE) as follows:

$$
\begin{aligned}
MAE &= \sum_{k=1}^{K} |\boldsymbol{p}(k|\boldsymbol{x}) - \boldsymbol{q}(k|\boldsymbol{x})| \\
&= (1 - \boldsymbol{p}(y|\boldsymbol{x})) + \sum_{k \neq y} \boldsymbol{p}(k|\boldsymbol{x}) \\
&= 2(1 - \boldsymbol{p}(y|\boldsymbol{x})),
\end{aligned}
\tag{16}
$$

where the last equality holds due to $\sum_{k=1}^{K} \boldsymbol{p}(k|\boldsymbol{x}) = 1$.

For Normalized Mean Absolute Error (NMAE), we have:

$$
\begin{aligned}
NMAE &= \frac{\sum_{k=1}^{K} |\boldsymbol{p}(k|\boldsymbol{x}) - \boldsymbol{q}(k|\boldsymbol{x})|}{\sum_{j=1}^{K} \sum_{k=1}^{K} |\boldsymbol{p}(k|\boldsymbol{x}) - \boldsymbol{q}(y = j|\boldsymbol{x})|} \\
&= \frac{2(1 - \boldsymbol{p}(y|\boldsymbol{x}))}{\sum_{j=1}^{K} 2(1 - \boldsymbol{p}(j|\boldsymbol{x}))} \\
&= \frac{1}{K-1}(1 - \boldsymbol{p}(y|\boldsymbol{x})) \\
&= \frac{1}{2(K-1)} \cdot MAE.
\end{aligned}
\tag{17}
$$

This shows that the NMAE is a scaled version of the MAE with a factor of $\frac{1}{2(K-1)}$.

For Reverse Cross Entropy (RCE), we have:

$$
\begin{aligned}
RCE &= \sum_{k=1}^{K} \boldsymbol{p}(x|\boldsymbol{x})(-\log \boldsymbol{q}(k|\boldsymbol{x})) \\
&= \boldsymbol{p}(y|\boldsymbol{x})(-\log 1) + \sum_{k \neq y} \boldsymbol{p}(k|\boldsymbol{x})(-\log 0) \\
&= \sum_{k \neq y} \boldsymbol{p}(k|\boldsymbol{x})(-A) \\
&= -A(1 - \boldsymbol{p}(y|\boldsymbol{x})) \\
&= -\frac{A}{2} \cdot MAE,
\end{aligned}
\tag{18}
$$

recall that in RCE we set $\log 0 = A$, where $A < 0$ is some constant (e.g., $A = -4$). This shows that the RCE is a scaled version of the MAE with a factor of $-\frac{A}{2}$.

For Normalized Reverse Cross Entropy (NRCE), we have:

$$
\begin{aligned}
NRCE &= \frac{\sum_{k=1}^{K} \boldsymbol{p}(k|\boldsymbol{x})(-\log \boldsymbol{q}(k|\boldsymbol{x}))}{\sum_{j=1}^{K} \sum_{k=1}^{K} \boldsymbol{p}(k|\boldsymbol{x})(-\log \boldsymbol{q}(y = j|\boldsymbol{x}))} \\
&= \frac{-A(1 - \boldsymbol{p}(y|\boldsymbol{x}))}{\sum_{j=1}^{K}(-A(1 - \boldsymbol{p}(j|\boldsymbol{x})))} \\
&= \frac{1}{K-1}(1 - \boldsymbol{p}(y|\boldsymbol{x})) \\
&= \frac{1}{2(K-1)} \cdot MAE.
\end{aligned}
\tag{19}
$$

This shows that NRCE is a scaled version of the MAE with a factor of $\frac{1}{2(K-1)}$.

### A.2 Normalized Negative Loss Functions

Here, we show how to derive Normalized Negative Loss Functions (NNLFs) into its proper form.

$$
\begin{aligned}
NNLF &= \frac{\sum_{k=1}^{K}(1 - \boldsymbol{q}(k|\boldsymbol{x}))(A - \mathcal{L}(f(\boldsymbol{x}), k))}{\sum_{j=1}^{K}\sum_{k=1}^{K}(1 - \boldsymbol{q}(k = j|\boldsymbol{x}))(A - \mathcal{L}(f(\boldsymbol{x}), k))} \\
&= \frac{\sum_{k \neq y} A - \mathcal{L}(f(\boldsymbol{x}), k)}{\sum_{j=1}^{K}\sum_{k \neq j} A - \mathcal{L}(f(\boldsymbol{x}), k)} \\
&= \frac{\sum_{k \neq y} A - \mathcal{L}(f(\boldsymbol{x}), k)}{(K - 1)\sum_{k=1}^{K} A - \mathcal{L}(f(\boldsymbol{x}), k)} \\
&= \frac{1}{K - 1} \cdot \left(1 - \frac{A - \mathcal{L}(f(\boldsymbol{x}), y)}{\sum_{k=1}^{K} A - \mathcal{L}(f(\boldsymbol{x}), k)}\right) \\
&\propto 1 - \frac{A - \mathcal{L}(f(\boldsymbol{x}), y)}{\sum_{k=1}^{K} A - \mathcal{L}(f(\boldsymbol{x}), k)}.
\end{aligned}
\tag{20}
$$

## B  Noise tolerant

**Theorem 1.** *Normalized negative loss function $\mathcal{L}_{nn}$ is symmetric.*

*Proof.* For all $\boldsymbol{x} \in \mathcal{X}$ and all $f$, we have:

$$
\begin{aligned}
\sum_{k=1}^{K} \mathcal{L}_{\mathrm{nn}}(f(\boldsymbol{x}), k) &= \sum_{k=1}^{K}\left(1 - \frac{A - \mathcal{L}(f(\boldsymbol{x}), k)}{\sum_{j=1}^{K} A - \mathcal{L}(f(\boldsymbol{x}), j)}\right) \\
&= K - \frac{\sum_{k=1}^{K} A - \mathcal{L}(f(\boldsymbol{x}), k)}{\sum_{j=1}^{K} A - \mathcal{L}(f(\boldsymbol{x}), j)} \\
&= K - 1,
\end{aligned}
\tag{21}
$$

where $K - 1$ is a constant and $\mathcal{L}_{\mathrm{nn}}$ satisfies the definition of the symmetric loss function. $\qquad\square$

## C  Gradient analysis

### C.1  Gradient of MAE

The complete derivation of the Mean Absolute Error (MAE) with respect to the classifier's output probabilities is as follows:

In the case of $j \neq y$:

$$
\begin{aligned}
\frac{\partial \mathcal{L}_{\mathrm{MAE}}}{\partial \boldsymbol{p}(j|\boldsymbol{x})} &= \frac{\partial}{\partial \boldsymbol{p}(j|\boldsymbol{x})} \sum_{k=1}^{K} |\boldsymbol{p}(k|\boldsymbol{x}) - \boldsymbol{q}(k|\boldsymbol{x})| \\
&= \frac{1}{\partial \boldsymbol{p}(j|\boldsymbol{x})}\left((1 - \boldsymbol{p}(y|\boldsymbol{x})) + \sum_{k \neq y} \boldsymbol{p}(k|\boldsymbol{x})\right) \\
&= 1.
\end{aligned}
\tag{22}
$$

In the case of $j = y$:

$$
\begin{aligned}
\frac{\partial \mathcal{L}_{\mathrm{MAE}}}{\partial \boldsymbol{p}(j|\boldsymbol{x})} &= \frac{\partial}{\partial \boldsymbol{p}(j|\boldsymbol{x})} \sum_{k=1}^{K} |\boldsymbol{p}(k|\boldsymbol{x}) - \boldsymbol{q}(k|\boldsymbol{x})| \\
&= \frac{1}{\partial \boldsymbol{p}(j|\boldsymbol{x})}\left((1 - \boldsymbol{p}(y|\boldsymbol{x})) + \sum_{k \neq y} \boldsymbol{p}(k|\boldsymbol{x})\right) \\
&= -1.
\end{aligned}
\tag{23}
$$

## C.2 Gradient of NNCE

The complete derivation of our proposed Normalized Negative Cross Entropy (NNCE) with respect to the classifier's output probabilities is as follows:

In the case of $j \neq y$:

$$
\begin{aligned}
\frac{\partial \mathcal{L}_{\text{NNCE}}}{\partial \boldsymbol{p}(j|\boldsymbol{x})} &= \frac{\partial}{\partial \boldsymbol{p}(j|\boldsymbol{x})} \left( 1 - \frac{A - (-\log \boldsymbol{p}(y|\boldsymbol{x}))}{\sum_{k=1}^{K} A - (-\log \boldsymbol{p}(k|\boldsymbol{x}))} \right) \\
&= -\frac{\partial}{\partial \boldsymbol{p}(j|\boldsymbol{x})} \left( \frac{A + \log \boldsymbol{p}(y|\boldsymbol{x})}{\sum_{k=1}^{K} A + \log \boldsymbol{p}(k|\boldsymbol{x})} \right) \\
&= -\frac{0 - (A + \log \boldsymbol{p}(y|\boldsymbol{x})) \frac{1}{\boldsymbol{p}(j|\boldsymbol{x})}}{\left( \sum_{k=1}^{K} A + \log \boldsymbol{p}(k|\boldsymbol{x}) \right)^2} \\
&= \frac{1}{\boldsymbol{p}(j|\boldsymbol{x})} \cdot \frac{A + \log \boldsymbol{p}(y|\boldsymbol{x})}{\left( \sum_{k=1}^{K} A + \log \boldsymbol{p}(k|\boldsymbol{x}) \right)^2}
\end{aligned}
\tag{24}
$$

In the case of $j = y$:

$$
\begin{aligned}
\frac{\partial \mathcal{L}_{\text{NNCE}}}{\partial \boldsymbol{p}(j|\boldsymbol{x})} &= \frac{\partial}{\partial \boldsymbol{p}(j|\boldsymbol{x})} \left( 1 - \frac{A - (-\log \boldsymbol{p}(y|\boldsymbol{x}))}{\sum_{k=1}^{K} A - (-\log \boldsymbol{p}(k|\boldsymbol{x}))} \right) \\
&= -\frac{\partial}{\partial \boldsymbol{p}(j|\boldsymbol{x})} \left( \frac{A + \log \boldsymbol{p}(y|\boldsymbol{x})}{\sum_{k=1}^{K} A + \log \boldsymbol{p}(k|\boldsymbol{x})} \right) \\
&= -\frac{\frac{1}{\boldsymbol{p}(y|\boldsymbol{x})} \left( \sum_{k=1}^{K} A + \log \boldsymbol{p}(k|\boldsymbol{x}) \right) - (A + \log \boldsymbol{p}(y|\boldsymbol{x})) \frac{1}{\boldsymbol{p}(y|\boldsymbol{x})}}{\left( \sum_{k=1}^{K} A + \log \boldsymbol{p}(k|\boldsymbol{x}) \right)^2} \\
&= -\frac{1}{\boldsymbol{p}(y|\boldsymbol{x})} \cdot \frac{\sum_{k \neq y} A + \log \boldsymbol{p}(k|\boldsymbol{x})}{\left( \sum_{k=1}^{K} A + \log \boldsymbol{p}(k|\boldsymbol{x}) \right)^2}
\end{aligned}
\tag{25}
$$

## C.3 Properties of NNCE

**Theorem 2.** *Given the classifier's output probability $\boldsymbol{p}(\cdot|\boldsymbol{x})$ for sample $\boldsymbol{x}$ and normalized negative cross entropy $\mathcal{L}_{\text{NNCE}}$. If $\boldsymbol{p}(j_1|\boldsymbol{x}) < \boldsymbol{p}(j_2|\boldsymbol{x})$, $j_1 \neq j_2 \neq y$, then $\frac{\partial \mathcal{L}_{\text{NNCE}}}{\partial \boldsymbol{p}(j_1|\boldsymbol{x})} > \frac{\partial \mathcal{L}_{\text{NNCE}}}{\partial \boldsymbol{p}(j_2|\boldsymbol{x})}$.*

*Proof.* As per the assumption, we have:

$$
\begin{aligned}
\boldsymbol{p}(j_1|\boldsymbol{x}) &< \boldsymbol{p}(j_2|\boldsymbol{x}) \\
\frac{1}{\boldsymbol{p}(j_1|\boldsymbol{x})} &> \frac{1}{\boldsymbol{p}(j_2|\boldsymbol{x})} \\
\frac{1}{\boldsymbol{p}(j_1|\boldsymbol{x})} \cdot \frac{A + \log \boldsymbol{p}(y|\boldsymbol{x})}{\left( \sum_{k=1}^{K} A + \log \boldsymbol{p}(k|\boldsymbol{x}) \right)^2} &> \frac{1}{\boldsymbol{p}(j_2|\boldsymbol{x})} \cdot \frac{A + \log \boldsymbol{p}(y|\boldsymbol{x})}{\left( \sum_{k=1}^{K} A + \log \boldsymbol{p}(k|\boldsymbol{x}) \right)^2} \\
\frac{\partial \mathcal{L}_{\text{NNCE}}}{\partial \boldsymbol{p}(j_1|\boldsymbol{x})} &> \frac{\partial \mathcal{L}_{\text{NNCE}}}{\partial \boldsymbol{p}(j_2|\boldsymbol{x})}.
\end{aligned}
\tag{26}
$$

This completes the proof. $\qquad\square$

**Theorem 3.** *Given the classifier's output probabilities $\boldsymbol{p}(\cdot|\boldsymbol{x}_1)$ and $\boldsymbol{p}(\cdot|\boldsymbol{x}_2)$ of sample $\boldsymbol{x}_1$ and $\boldsymbol{x}_2$, where $\boldsymbol{p}(y|\boldsymbol{x}_1) \geq \boldsymbol{p}(k|\boldsymbol{x}_1)$, $\boldsymbol{p}(y|\boldsymbol{x}_2) \geq \boldsymbol{p}(k|\boldsymbol{x}_2)$, $\forall k \in \{1, \cdots, K\}$, $\boldsymbol{p}(j|\boldsymbol{x}_1) = \boldsymbol{p}(j|\boldsymbol{x}_2)$, and normalized negative cross entropy $\mathcal{L}_{\text{NNCE}}$. If $\boldsymbol{p}(y|\boldsymbol{x}_1) > \boldsymbol{p}(y|\boldsymbol{x}_2)$, $j \neq y$, and $\boldsymbol{p}(k|\boldsymbol{x}_1) \leq \boldsymbol{p}(k|\boldsymbol{x}_2)$, $\forall k \in \{1, \cdots, K\} \setminus \{j, y\}$, then $\frac{\partial \mathcal{L}_{\text{NNCE}}}{\partial \boldsymbol{p}(j|\boldsymbol{x}_1)} > \frac{\partial \mathcal{L}_{\text{NNCE}}}{\partial \boldsymbol{p}(j|\boldsymbol{x}_2)}$.*

*Proof.* We first consider two functions $f_1$ and $f_2$. The function $f_1$ is defined as follows:

$$
f_1(\boldsymbol{p}) = \sum_{k=1}^{K} \log \boldsymbol{p}_k,
\tag{27}
$$

where the input $\boldsymbol{p}$ is a discrete probability distribution, $\sum_{k=1}^{K} \boldsymbol{p}_k = 1$ and $0 \leq \boldsymbol{p}_k \leq 1$, $\forall k \in \{1, \cdots, K\}$. Given a discrete probability distribution $\boldsymbol{p}$ and a real number $D$ as input, the function $f_2$ is defined as follows:

$$f_2(\boldsymbol{p}, D) = \log(\boldsymbol{p}_y + D) + \log \boldsymbol{p}_j + \sum_{k \neq j \neq y} \log(\boldsymbol{p}_k - d_k), \tag{28}$$

where $\boldsymbol{p}_y \geq \boldsymbol{p}_k$, $\forall k \in \{1, \cdots, K\}$, $0 < D \leq 1 - \boldsymbol{p}_y$. And $\{d_k\}$, $k \in \{1, \cdots, K\} \setminus \{j, y\}$ is a set of random variables which satisfy following conditions: $0 \leq \boldsymbol{p}_k - d_k \leq \boldsymbol{p}_k$ and $\sum_{k \neq j \neq y} d_k = D$.

Next, we consider whether $f_1(\boldsymbol{p}) - f_2(\boldsymbol{p}, D) > 0$. Given $\boldsymbol{p}$ and $D$, we have,

$$
\begin{aligned}
f_1(\boldsymbol{p}) - f_2(\boldsymbol{p}, D) &= \Big( \sum_{k=1}^{K} \log \boldsymbol{p}_k \Big) - \Big( \log(\boldsymbol{p}_y + D) + \log \boldsymbol{p}_j + \sum_{k \neq j \neq y} \log(\boldsymbol{p}_k - d_k) \Big) \\
&= \log \boldsymbol{p}_y - \log(\boldsymbol{p}_y + D) + \sum_{k \neq j \neq y} \log \boldsymbol{p}_k - \log(\boldsymbol{p}_k - d_k) \\
&= \log \frac{\boldsymbol{p}_y}{\boldsymbol{p}_y + D} + \sum_{k \neq j \neq y} \log \frac{\boldsymbol{p}_k}{\boldsymbol{p}_k - d_k} \\
&\geq \log \frac{\boldsymbol{p}_y}{\boldsymbol{p}_y + D} + \sum_{k \neq j \neq y} \log \frac{\boldsymbol{p}_y}{\boldsymbol{p}_y - d_k} \\
&= \log \boldsymbol{p}_y - \log(\boldsymbol{p}_y + D) + \sum_{k \neq j \neq y} \log \boldsymbol{p}_y - \log(\boldsymbol{p}_y - d_k) \\
&= \Big( \log \boldsymbol{p}_y + \sum_{k \neq j \neq y} \log \boldsymbol{p}_y \Big) - \Big( \log(\boldsymbol{p}_y + D) + \sum_{k \neq j \neq y} \log(\boldsymbol{p}_y - d_k) \Big) \\
&\geq \Big( \log \boldsymbol{p}_y + \sum_{k \neq j \neq y} \log \boldsymbol{p}_y \Big) - \sup_{\{d_k\}, k \neq j \neq y} \Big( \log(\boldsymbol{p}_y + D) + \sum_{k \neq j \neq y} \log(\boldsymbol{p}_y - d_k) \Big).
\end{aligned}
\tag{29}
$$

The first inequality holds because for a fraction $\frac{a}{b} \geq 1$, we always have $\frac{a}{b} \geq \frac{a+c}{b+c}$, where $c \geq 0$.

To get the maximum value of $\log(\boldsymbol{p}_y + D) + \sum_{k \neq j \neq y} \log(\boldsymbol{p}_y - d_k)$, we must solve the following minimization problem subject to constraints:

$$\min_{\{d_k\}, k \neq j \neq y} \quad -\Big( \log(\boldsymbol{p}_y + D) + \sum_{k \neq j \neq y} \log(\boldsymbol{p}_y - d_k) \Big). \tag{30}$$

$$\text{s.t.} \quad \sum_{k \neq j \neq y} d_k = D. \tag{31}$$

We can define the Lagrange function $L$ as follows:

$$L(d_1, \cdots, d_K, \lambda) = -\Big( \log(\boldsymbol{p}_y + D) + \sum_{k \neq j \neq y} \log(\boldsymbol{p}_y + d_k) \Big) + \lambda \cdot \Big( \sum_{k \neq j \neq y} d_k - D \Big). \tag{32}$$

Now we can calculate the gradients:

$$\begin{cases} \frac{\partial L}{\partial d_k} = -\frac{1}{\boldsymbol{p}_y + d_k} + \lambda, k \neq j \neq y \\ \frac{\partial L}{\partial \lambda} = \sum_{k \neq j \neq y} d_k - D. \end{cases} \tag{33}$$

Let $\frac{\partial L}{\partial d_k} = 0$ and $\frac{\partial L}{\partial \lambda} = 0$, we have:

$$\begin{cases} -\frac{1}{\boldsymbol{p}_y + d_k} + \lambda = 0, k \neq j \neq y \\ \sum_{k \neq j \neq y} d_k - D = 0, \end{cases} \tag{34}$$

and therefore:

$$d_k = \frac{D}{K - 2}, k \neq j \neq y. \tag{35}$$

So, the minimization value is:

$$-\Big( \log(\boldsymbol{p}_y + D) + \sum_{k \neq j \neq y} \log(\boldsymbol{p}_y - \frac{D}{K-2}) \Big). \tag{36}$$

Now, back to the eq. (29), we have:

$$f_1(\boldsymbol{p}) - f_2(\boldsymbol{p}, D) \geq \log \boldsymbol{p}_y + \sum_{k \neq j \neq y} \log \boldsymbol{p}_y - \sup_{\{d_k\}, k \neq j \neq y} \Big( \log(\boldsymbol{p}_y + D) + \sum_{k \neq j \neq y} \log(\boldsymbol{p}_y - d_k) \Big)$$

$$= \log \boldsymbol{p}_y + \sum_{k \neq j \neq y} \log \boldsymbol{p}_y - \Big( \log(\boldsymbol{p}_y + D) + \sum_{k \neq j \neq y} \log(\boldsymbol{p}_y - \frac{D}{K-2}) \Big)$$

$$= \log \boldsymbol{p}_y - \log(\boldsymbol{p}_y + D) + (K-2)\Big( \log \boldsymbol{p}_y - \log(\boldsymbol{p}_y - \frac{D}{K-2}) \Big)$$

$$= (K-2)\Big( \log \boldsymbol{p}_y - \log(\boldsymbol{p}_y - \frac{D}{K-2}) \Big) - \Big( \log(\boldsymbol{p}_y + D) - \log \boldsymbol{p}_y \Big)$$

$$= D \cdot \Big( \frac{\log \boldsymbol{p}_y - \log(\boldsymbol{p}_y - \frac{D}{K-2})}{\frac{D}{K-2}} - \frac{\log(\boldsymbol{p}_y + D) - \log \boldsymbol{p}_y}{D} \Big). \tag{37}$$

Recall the nature of the difference, we have $\frac{\log \boldsymbol{p}_y - \log(\boldsymbol{p}_y - \frac{D}{K-2})}{\frac{D}{K-2}} = \frac{d \log x}{dx}\Big|_{x = \boldsymbol{p}_y - \alpha} = \frac{1}{\boldsymbol{p}_y - \alpha}$, where $0 \leq \alpha \leq \frac{D}{K-2}$, and $\frac{\log(\boldsymbol{p}_y + D) - \log \boldsymbol{p}_y}{D} = \frac{d \log x}{dx}\Big|_{x = \boldsymbol{p}_y + \beta} = \frac{1}{\boldsymbol{p}_y + \beta}$, where $0 \leq \beta \leq D$. And therefore:

$$\frac{\log \boldsymbol{p}_y - \log(\boldsymbol{p}_y - \frac{D}{K-2})}{\frac{D}{K-2}} = \frac{1}{\boldsymbol{p}_y - \alpha} > \frac{1}{\boldsymbol{p}_y} > \frac{1}{\boldsymbol{p}_y + \beta} = \frac{\log(\boldsymbol{p}_y + D) - \log \boldsymbol{p}_y}{D}. \tag{38}$$

So we have:

$$\frac{\log \boldsymbol{p}_y - \log(\boldsymbol{p}_y - \frac{D}{K-2})}{\frac{D}{K-2}} > \frac{\log(\boldsymbol{p}_y + D) - \log \boldsymbol{p}_y}{D} \tag{39}$$

$$f_1(\boldsymbol{p}) > f_2(\boldsymbol{p}, D).$$

Now, let $\boldsymbol{p}_k = \boldsymbol{p}(k|\boldsymbol{x}_2), \forall k \in \{1, \cdots, K\}$, $D = \boldsymbol{p}(y|\boldsymbol{x}_1) - \boldsymbol{p}(y|\boldsymbol{x}_2)$, and $d_k = \boldsymbol{p}(k|\boldsymbol{x}_2) - \boldsymbol{p}(k|\boldsymbol{x}_1)$, $k \neq j \neq y$. Following eq. (39), we have:

$$f_1(\boldsymbol{p}) > f_2(\boldsymbol{p}, D)$$

$$\sum_{k=1}^{K} \log \boldsymbol{p}_k > \log(\boldsymbol{p}_y + D) + \log \boldsymbol{p}_j + \sum_{k \neq j \neq y} \log(\boldsymbol{p}_k - d_k)$$

$$\sum_{k=1}^{K} \log \boldsymbol{p}(k|\boldsymbol{x}_2) > \sum_{k=1}^{K} \log \boldsymbol{p}(k|\boldsymbol{x}_1)$$

$$\frac{1}{\Big( \sum_{k=1}^{K} A + \log \boldsymbol{p}(k|\boldsymbol{x}_1) \Big)^2} > \frac{1}{\Big( \sum_{k=1}^{K} A + \log \boldsymbol{p}(k|\boldsymbol{x}_2) \Big)^2} \tag{40}$$

$$\frac{1}{\boldsymbol{p}(j|\boldsymbol{x}_1)} \cdot \frac{A + \log \boldsymbol{p}(y|\boldsymbol{x}_1)}{\Big( \sum_{k=1}^{K} A + \log \boldsymbol{p}(k|\boldsymbol{x}_1) \Big)^2} > \frac{1}{\boldsymbol{p}(j|\boldsymbol{x}_2)} \cdot \frac{A + \log \boldsymbol{p}(y|\boldsymbol{x}_2)}{\Big( \sum_{k=1}^{K} A + \log \boldsymbol{p}(k|\boldsymbol{x}_2) \Big)^2}$$

$$\frac{\partial \mathcal{L}_{\text{NNCE}}}{\partial \boldsymbol{p}(j|\boldsymbol{x}_1)} > \frac{\partial \mathcal{L}_{\text{NNCE}}}{\partial \boldsymbol{p}(j|\boldsymbol{x}_2)}.$$

The eq. (40) holds because we have $\boldsymbol{p}(j|\boldsymbol{x}_1) = \boldsymbol{p}(j|\boldsymbol{x}_2)$, and $\log \boldsymbol{p}(y|\boldsymbol{x}_1) > \log \boldsymbol{p}(y|\boldsymbol{x}_2)$, where $\boldsymbol{p}(y|\boldsymbol{x}_1) > \boldsymbol{p}(y|\boldsymbol{x}_2)$. This completes the proof. $\qquad \square$

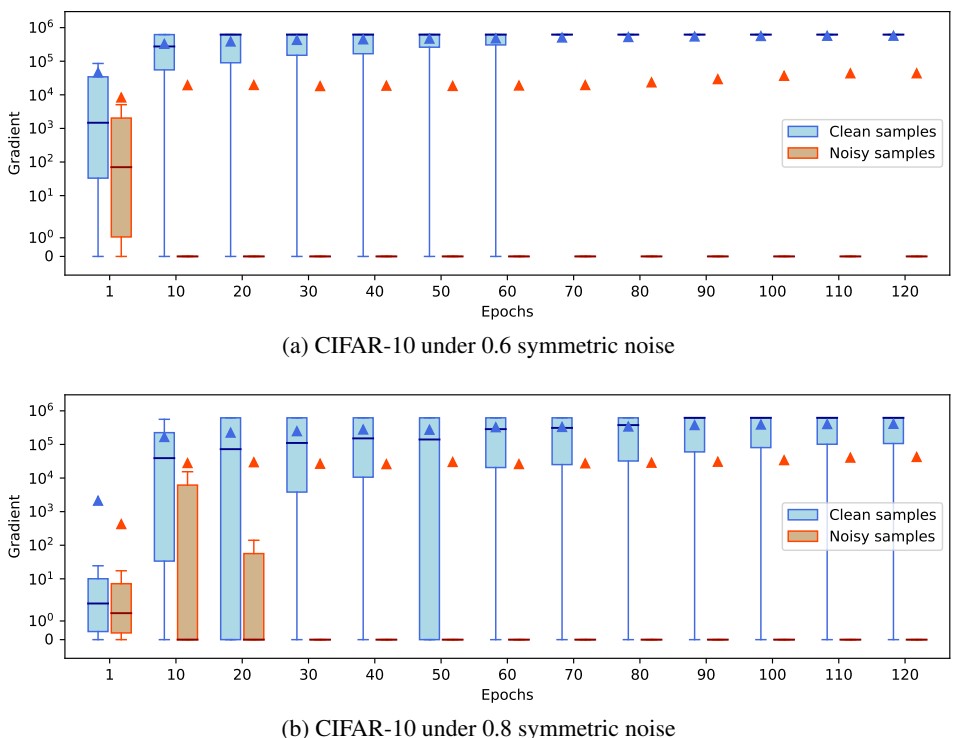

(a) CIFAR-10 under 0.6 symmetric noise

(b) CIFAR-10 under 0.8 symmetric noise

Figure 4: The gradient $\max_{j,j\neq y}\left(\frac{\partial \mathcal{L}_{\mathrm{NNCE}}}{\partial \boldsymbol{p}(j|\boldsymbol{x})}\right)$ for clean and noisy samples. Blue boxes indicate clean samples, red boxes indicate noisy samples, blue triangles indicate the means of clean samples, and red triangles indicate the means of noisy samples. The outliers are ignored.

# D   Experiments

## D.1   Empirical Understandings

**Gradients of NNCE.** We conduct experiments with ANL-CE on CIFAR-10 under 0.6 and 0.8 symmetric noise. For each sample, we calculate the gradient of our NNCE with respect to the predicted probability of each non-labeled class and take their maximum value, which is $\max_{j,j\neq y}\left(\frac{\partial \mathcal{L}_{\mathrm{NNCE}}}{\partial \boldsymbol{p}(j|\boldsymbol{x})}\right)$. The results are shown in fig. 4 in the form of box plots. As can be observed, as the number of training epochs increases, the gradients of the clean samples are concentrated at larger values and the gradients of the noisy samples are concentrated at smaller values. The means of the gradients for noisy samples are higher than the medians and smaller than the means for clean samples' gradients. This indicates that although the model is still incorrectly learning some noisy samples, these learned noisy samples constitute only a small fraction of all noisy samples in the training set. In general, this result verifies that our NNLFs have the ability to focus more on clean samples and ignore noisy samples during the training process, and also verifies our discussion in section 3.3.

**Parameter Analysis.** We apply different values of $\alpha$ and $\beta$ to NCE+NNCE for training on CIFAR-10 under 0.8 symmetric noise. We test the combinations between $\alpha \in \{0.1, 0.5, 1.0, 5.0, 10.0\}$ and $\beta \in \{0.1, 0.5, 1.0, 5.0, 10.0\}$, where $\alpha$ is the weight of NCE and $\beta$ is the weight of NNCE. The weight decay of NCE+NNCE is set to be the same as NCE+RCE. As can be observed in fig. 5, regardless of how $\alpha$ (the weight of NCE) varies, when $\beta$ (the weight of NNCE) increases, both the robustness and the fitting ability of the model improve, although overfitting occurs. This verifies that our NNLFs are robust to noisy labels and have good fitting ability. Although in ANL we use L1 regularization instead of L2 regularization (weight decay), the effect of $\alpha$ and $\beta$ in the loss functions created by ANL should be similar to that in NCE+NNCE.

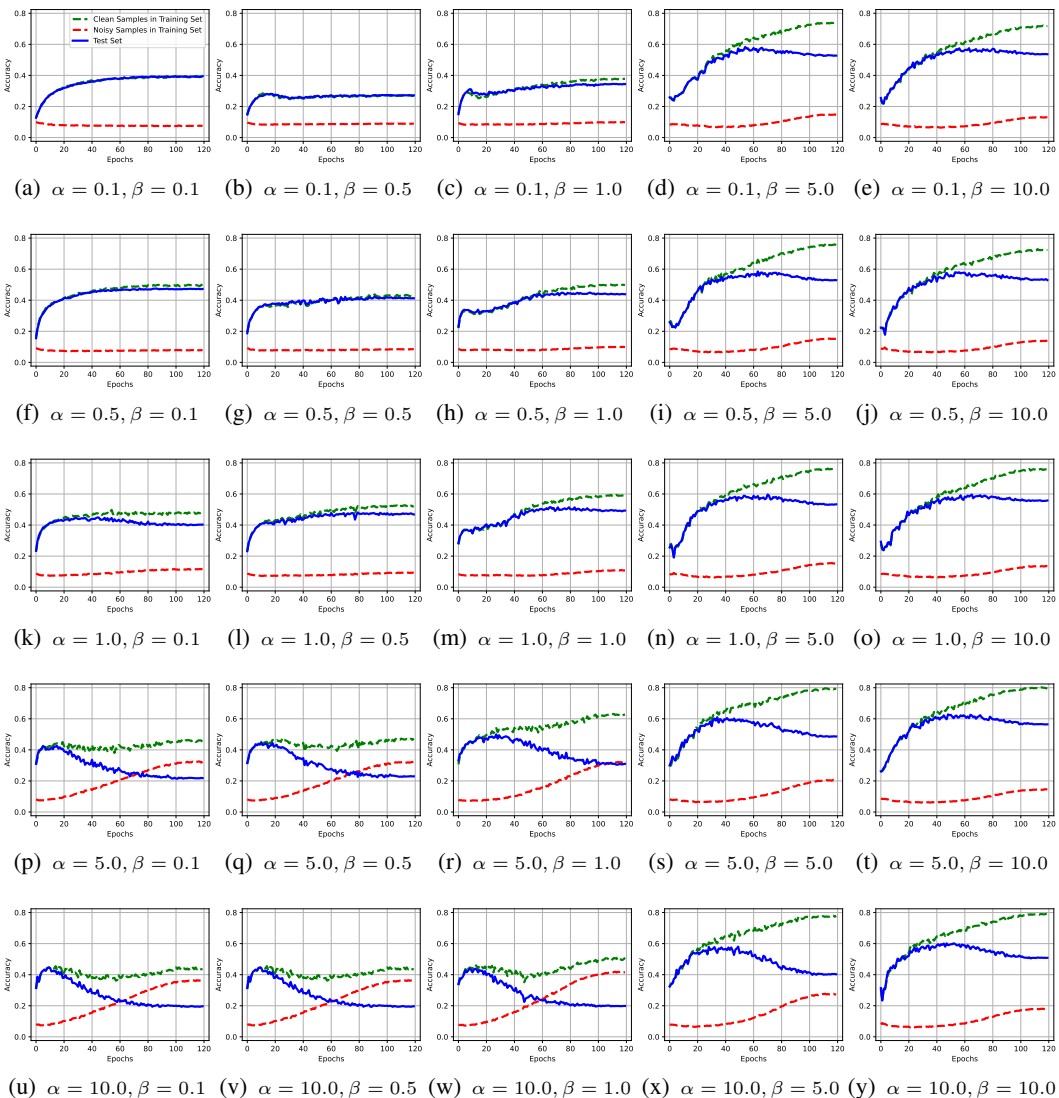

Figure 5: Training and test accuracies of NCE+NNCE on CIFAR-10 under 0.8 symmetric noise with different parameters. $\alpha$ is the weight of NCE and $\beta$ is the weight of NNCE. The accuracies of noisy samples in training set (red dashed line) should be as low as possible, since they are mislabeled.

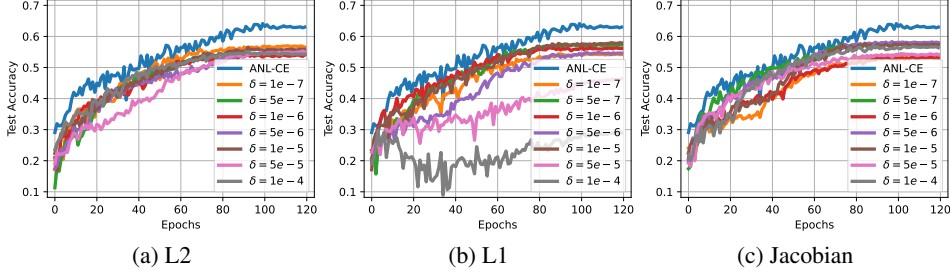

Figure 6: Test accuracies of NCE+RCE on CIFAR-10 under 0.8 symmetric noise with different regularization methods and different parameters. $\delta$ is the weight of regularization term of NCE+RCE. As a comparison, the blue line shows the test accuracy of our ANL-CE with the same data set and noise setting.

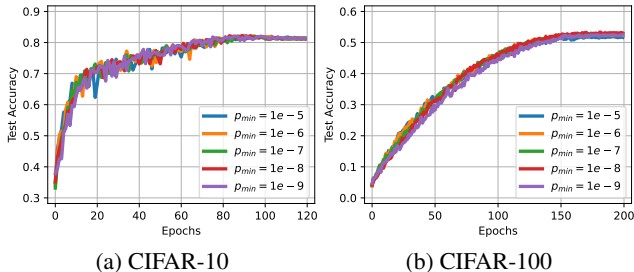

(a) CIFAR-10          (b) CIFAR-100

Figure 7: Test accuracies of different $p_{\min}$ applied to ANL-CE on CIFAR-10/CIFAR-100 under 0.6 symmetric noise.

**Can APL outperform our ANL by changing the regularization method?** We apply different regularization methods to NCE+RCE for training on CIFAR-10 under 0.8 symmetric noise and compare the results with our ANL-CE. Specifically, we train networks using loss functions in the form of $\alpha \cdot NCE + \beta \cdot RCE + \delta \cdot \text{Reg}$, where $\alpha, \beta, \delta > 0$ are parameters and Reg is the regularization term. Similarly, we consider 3 regularization methods: 1) L2, 2) L1, and 3) Jacobian [15, 16]. We keep the $\alpha$ and $\beta$ of NCE+RCE the same as in section 4.2 and tune $\delta \in \{ 1 \times 10^{-7}, 5 \times 10^{-7}, 1 \times 10^{-6} \ 5 \times 10^{-6} \ 1 \times 10^{-5} \ 5 \times 10^{-5} \ 1 \times 10^{-4} \}$ for all three methods. As can be observed in fig. 6, the performance of NCE+RCE can be improved by adapting the regularization method. However, the improvement from changing the regularization method is limited, and the test accuracies of NCE+RCE are consistently lower than our ANL-CE, no matter how the regularization method is changed and how the parameter $\delta$ is varied. This verifies that APL cannot outperform our ANL by changing the regularization method and also verifies the performance of our proposed NNLFs.

**Does the $p_{\min}$ affect the performance of ANL?** As shown in the definitions of NLF and NNLF, $p_{\min}$ controls the lower bound of the classifier's output probability, and further controls the constant $A$. In fig. 7, we show different $p_{\min} \in \{1 \times 10^{-5}, 1 \times 10^{-6}, 1 \times 10^{-7}, 1 \times 10^{-8}, 1 \times 10^{-9}\}$ applied to ANL-CE for training on CIFAR-10/-100 under 0.6 symmetric noise. As can be observed, for both datasets, $p_{\min}$ does not significant affect the performance of ANL.

## D.2   Evaluation on Benchmark Datasets

**Noise generation.** The noisy labels are generated following standard approaches in previous works [28, 6, 17]. For symmetric noise, we flip the labels in each class randomly to incorrect labels of other classes. For asymmetric noise, we flip the labels within a specific set of classes. For MNIST, flipping $7 \rightarrow 1$, $2 \rightarrow 7$, $5 \leftrightarrow 6$, $3 \rightarrow 8$. For CIFAR-10, flipping TRUCK $\rightarrow$ AUTOMOBILE, BIRD $\rightarrow$ AIRPLANE, DEER $\rightarrow$ HORSE, CAT $\leftrightarrow$ DOG. For CIFAR-100, the 100 classes are grouped into 20 super-classes with each has 5 sub-classes, and each class are flipped within the same super-class into the next in a circular fashion. We vary the noise rate $\eta \in \{0.2, 0.4, 0.6, 0.8\}$ for symmetric noise and $\eta \in \{0.1, 0.2, 0.3, 0.4\}$ for asymmetric noise.

**Networks and training.** We follow the experimental setting in previous works [6, 17]. A 4-layer CNN is used for MNIST, an 8-layer CNN is used for CIFAR-10, and a ResNet-34 [29] is used for CIFAR-100. For MNIST, CIFAR-10, and CIFAR-100, the networks are trained for 50, 120, and 200 epochs, respectively. For all the training, we use SGD optimizer with momentum 0.9 and cosine learning rate annealing. Weight decay is set to $1 \times 10^{-3}$, $1 \times 10^{-4}$, and $1 \times 10^{-5}$ for MNIST, CIFAR-10, and CIFAR-100, respectively. Particularly, for our proposed ANL methods, weight decay is set to 0 for all datasets. The initial learning rate is set to 0.01 for MNIST/CIFAR-10 and 0.1 for CIFAR-100. Batch size is set to 128. For all settings, we clip the gradient norm to 5.0. Typical data augmentations including random width/height shift and horizontal flip are applied.

**Parameter tuning.** For each dataset, we tune the parameters of ANL-CE under 0.8 symmetric noise and then use them directly for all other noise settings and ANL functions. Specifically, we use 10% of the original training set as the validation set, and generate 0.8 symmetric noise on the remaining 90% of the original training set as the training set by the standard noise generation approach. We tune the parameters $\alpha \in \{0.1, 0.5, 1.0, 5.0, 10.0\}$, $\beta \in \{0.1, 0.5, 1.0, 5.0, 10.0\}$, and $\delta \in \{ 1 \times 10^{-7}, 5 \times 10^{-7}, 1 \times 10^{-6}, 5 \times 10^{-6}, 1 \times 10^{-5}, 5 \times 10^{-5}, 1 \times 10^{-4} \}$.

Table 4: Parameters settings for different methods.

| Method | MNIST | CIFAR-10 | CIFAR-100 | WebVision |
|---|---|---|---|---|
| CE (-) | (-) | (-) | (-) | (-) |
| FL ($\gamma$) | (0.5) | (0.5) | (0.5) | - |
| MAE (-) | (-) | (-) | (-) | - |
| GCE ($q$) | (0.7) | (0.7) | (0.7) | (0.7) |
| SCE ($\alpha, \beta$) | (0.01, 1.0) | (0.1, 1.0) | (6.0, 0.1) | (10.0, 1.0) |
| NCE+MAE ($\alpha, \beta$) | (1.0, 10.0) | (1.0, 1.0) | (10.0, 0.1) | - |
| NCE+RCE ($\alpha, \beta$) | (1.0, 10.0) | (1.0, 1.0) | (10.0, 0.1) | (50.0, 0.1) |
| NFL+RCE ($\alpha, \beta, \gamma$) | (1.0, 10.0, 0.5) | (1.0, 1.0, 0.5) | (10.0, 0.1, 0.5) | - |
| NCE+AEL ($\alpha, \beta, a$) | (0.0, 1.0, 3.5) | (1.0, 4.0, 5.0) | (10.0, 0.1, 1.5) | - |
| NCE+AGCE ($\alpha, \beta, a, q$) | (0.0, 1.0, 4.0, 0.2) | (1.0, 4.0, 6.0, 1.5) | (10.0, 0.1, 1.8, 3.0) | (50.0, 0.1, 2.5, 3.0) |
| NCE+AUL ($\alpha, \beta, a, p$) | (0.0, 1.0, 3.0, 0.1) | (1.0, 4.0, 6.3, 1.5) | (10.0, 0.015, 6.0, 3.0) | - |
| ANL-CE ($\alpha, \beta, \delta$) | $(1.0, 1.0, 1 \times 10^{-6})$ | $(5.0, 5.0, 5 \times 10^{-5})$ | $(10.0, 1.0, 5 \times 10^{-7})$ | $(20.0, 1.0, 6 \times 10^{-6})$ |
| ANL-FL ($\alpha, \beta, \delta, \gamma$) | $(1.0, 1.0, 1 \times 10^{-6}, 0.5)$ | $(5.0, 5.0, 5 \times 10^{-5}, 0.5)$ | $(10.0, 1.0, 5 \times 10^{-7}, 0.5)$ | $(20.0, 1.0, 6 \times 10^{-6}, 0.5)$ |

Table 5: Test accuracies (%) of different methods on benchmark datasets with symmetric ($\eta \in \{0.2, 0.4, 0.6, 0.8\}$) or asymmetric ($\eta \in \{0.1, 0.2, 0.3, 0.4\}$) label noise. The results (mean±std) of our methods are reported over 3 random runs under different random seeds $(1, 2, 3)$. The reported results of PHuber-CE and Taylor-CE are directly taken from the original paper of Taylor-CE. The top-2 best results are in **bold**.

| Datasets | Methods | Symmetric Noise Rate ($\eta$) | | | | Asymmetric Noise Rate ($\eta$) | | | |
|---|---|---|---|---|---|---|---|---|---|
| | | 0.2 | 0.4 | 0.6 | 0.8 | 0.1 | 0.2 | 0.3 | 0.4 |
| CIFAR-10 | PHuber-CE [25] | 85.81±0.21 | 80.25±0.22 | 67.71±0.19 | 32.97±0.32 | 87.91±0.13 | 84.87±0.26 | 79.01±0.38 | 71.87±0.36 |
| | Taylor-CE [26] | 85.96±0.09 | 80.51±0.11 | 66.36±0.32 | 33.48±0.44 | 87.34±0.12 | 85.02±0.11 | 79.37±0.12 | 72.65±0.11 |
| | ANL-CE (ours) | **87.43±0.24** | **83.18±0.54** | **73.80±0.24** | 45.18±1.27 | **88.29±0.07** | **86.54±0.27** | **83.12±0.58** | **75.07±0.43** |
| | ANL-FL (ours) | 86.75±0.70 | 82.97±0.46 | 74.47±0.21 | 45.52±2.24 | 88.42±0.16 | 86.36±0.54 | 83.26±0.14 | 74.95±1.21 |
| CIFAR-100 | PHuber-CE [25] | 58.11±0.11 | 50.89±0.13 | 35.85±0.29 | 13.83±0.25 | 60.07±0.09 | 53.30±0.10 | 44.39±0.14 | 35.36±0.13 |
| | Taylor-CE [26] | 59.11±0.11 | 50.99±0.09 | 38.31±0.12 | 15.96±0.31 | 60.96±0.21 | 55.45±0.12 | 45.81±0.19 | 35.45±0.25 |
| | ANL-CE (ours) | **59.20±0.36** | 53.42±0.18 | 44.42±0.09 | 26.27±0.81 | 60.58±0.68 | **58.11±0.60** | 49.91±0.42 | 38.57±0.72 |
| | ANL-FL (ours) | **59.23±0.17** | 53.79±0.50 | 44.71±0.64 | 26.89±0.62 | **61.51±0.33** | 58.05±0.11 | 50.27±0.54 | 39.14±0.98 |

**Parameter settings.** For all baseline methods, the parameters are set to match their original papers. Detailed parameter settings can be found in table 4.

**Results.** The experimental results of symmetric and asymmetric label noise are reported in table 6 and table 7, respectively.

### D.3 Evaluation on Real-world Noisy Labels

We train a ResNet-50 [29] using SGD for 250 epochs with initial learning rate 0.4, nesterov momentum 0.9 and weight decay $3 \times 10^{-5}$ (for our proposed ANL methods, the weight decay is set to 0) and batch size 512. The learning rate is multiplied by 0.97 after every epoch of training. For all settings, we clip the gradient norm to 5.0. All the images are resized to $224 \times 224$. Typical data augmentations including random width/height shift, color jittering and random horizontal flip are applied. Detailed parameter settings can be found in table 4.

### D.4 More comparisons with other baseline methods

We also compare our proposed ANL with two other baseline methods, PHuber-CE [25] and Taylor-CE [26]. For a fair comparison, we followed the experimental setup in the Taylor-CE paper and compared it directly with its reported experimental results. We use Adam [30] optimizer with the number of epochs set to 200 and the batch size set to 256. We use ResNet-34 for CIFAR-10/-100. We tune our parameters and select the learning rate lr from $\{10^{-2}, 10^{-3}, 10^{-4}, 10^{-5}\}$. For CIFAR-10 and CIFAR-100, we set the parameters $(\alpha, \beta, \delta, \text{lr})$ to $(5.0, 5.0, 1 \times 10^{-7}, 1 \times 10^{-4})$ and $(10.0, 1.0, 1 \times 10^{-7}, 1 \times 10^{-4})$, respectively. The results in table 5 demonstrate that our proposed ANL can outperform both PHuber-CE and Taylor-CE on the CIFAR-10/-100 dataset with symmetric or asymmetric label noise.

Table 6: Test accuracies (%) of different methods on benchmark datasets with clean or symmetric label noise ($\eta \in \{0.2, 0.4, 0.6, 0.8\}$). The results (mean$\pm$std) are reported over 3 random runs under different random seeds $(1, 2, 3)$ and the top-2 best results are **boldfaced**.

| Datasets | Methods | Clean ($\eta$=0.0) | Symmetric Noise Rate ($\eta$) | | | |
| --- | --- | --- | --- | --- | --- | --- |
| | | | 0.2 | 0.4 | 0.6 | 0.8 |
| MNIST | CE | 99.20$\pm$0.02 | 91.40$\pm$0.28 | 74.46$\pm$0.28 | 49.19$\pm$0.05 | 22.51$\pm$0.23 |
| | FL | 99.16$\pm$0.08 | 91.66$\pm$0.18 | 75.42$\pm$0.25 | 50.58$\pm$0.38 | 22.93$\pm$0.11 |
| | MAE | 99.16$\pm$0.03 | 99.03$\pm$0.01 | 98.80$\pm$0.02 | 97.69$\pm$0.20 | 70.35$\pm$1.16 |
| | GCE | 99.18$\pm$0.01 | 98.84$\pm$0.03 | 96.81$\pm$0.13 | 80.86$\pm$0.31 | 33.59$\pm$0.48 |
| | SCE | 99.30$\pm$0.07 | 98.91$\pm$0.04 | 97.48$\pm$0.16 | 88.35$\pm$0.77 | 48.28$\pm$0.81 |
| | NLNL | 98.61$\pm$0.13 | 98.02$\pm$0.14 | 97.17$\pm$0.09 | 95.42$\pm$0.30 | 86.34$\pm$1.43 |
| | NCE+MAE | 99.22$\pm$0.06 | 98.96$\pm$0.04 | 98.15$\pm$0.14 | 92.94$\pm$0.26 | 59.54$\pm$0.76 |
| | NCE+RCE | 99.43$\pm$0.02 | **99.20$\pm$0.05** | 98.53$\pm$0.09 | 95.61$\pm$0.12 | 74.04$\pm$1.83 |
| | NFL+RCE | 99.33$\pm$0.04 | **99.17$\pm$0.03** | 98.62$\pm$0.03 | 95.54$\pm$0.26 | 75.05$\pm$1.45 |
| | NCE+AEL | 99.06$\pm$0.06 | 99.03$\pm$0.08 | 98.87$\pm$0.05 | 98.39$\pm$0.11 | 96.71$\pm$0.10 |
| | NCE+AGCE | 99.10$\pm$0.03 | 99.00$\pm$0.05 | **98.91$\pm$0.04** | **98.50$\pm$0.07** | **96.93$\pm$0.13** |
| | NCE+AUL | 99.19$\pm$0.01 | 99.04$\pm$0.05 | **99.00$\pm$0.03** | **98.52$\pm$0.14** | **96.97$\pm$0.12** |
| | **ANL-CE** | 99.08$\pm$0.05 | 98.97$\pm$0.02 | 98.84$\pm$0.05 | 98.42$\pm$0.08 | 96.62$\pm$0.12 |
| | **ANL-FL** | 99.13$\pm$0.05 | 98.94$\pm$0.07 | 98.90$\pm$0.05 | 98.46$\pm$0.12 | 95.73$\pm$0.22 |
| CIFAR-10 | CE | 90.38$\pm$0.11 | 75.05$\pm$0.26 | 58.19$\pm$0.21 | 38.75$\pm$0.19 | 19.09$\pm$0.35 |
| | FL | 89.84$\pm$0.28 | 74.52$\pm$0.10 | 57.54$\pm$0.75 | 38.83$\pm$0.49 | 19.33$\pm$0.58 |
| | MAE | 89.15$\pm$0.27 | 87.19$\pm$0.19 | 81.76$\pm$3.17 | 76.82$\pm$0.84 | 46.42$\pm$3.66 |
| | GCE | 89.66$\pm$0.20 | 87.17$\pm$0.01 | 82.44$\pm$0.26 | 68.62$\pm$0.35 | 25.45$\pm$0.51 |
| | SCE | 91.38$\pm$0.12 | 87.86$\pm$0.12 | 79.96$\pm$0.25 | 62.16$\pm$0.33 | 27.98$\pm$0.98 |
| | NLNL | 90.73$\pm$0.20 | 73.70$\pm$0.05 | 63.90$\pm$0.44 | 50.68$\pm$0.47 | 29.53$\pm$1.55 |
| | NCE+MAE | 88.94$\pm$0.13 | 87.37$\pm$0.19 | 83.70$\pm$0.21 | 76.35$\pm$0.08 | 44.68$\pm$1.12 |
| | NCE+RCE | 90.94$\pm$0.01 | 89.19$\pm$0.18 | 86.03$\pm$0.13 | 79.89$\pm$0.11 | 55.52$\pm$2.74 |
| | NFL+RCE | 90.86$\pm$0.51 | 89.04$\pm$0.09 | 86.08$\pm$0.33 | 79.79$\pm$0.16 | 54.18$\pm$2.06 |
| | NCE+AEL | 88.51$\pm$0.26 | 86.59$\pm$0.24 | 83.07$\pm$0.46 | 75.06$\pm$0.26 | 41.79$\pm$1.40 |
| | NCE+AGCE | 91.08$\pm$0.06 | 89.11$\pm$0.07 | 86.16$\pm$0.10 | 80.14$\pm$0.27 | 55.62$\pm$4.78 |
| | NCE+AUL | 91.26$\pm$0.12 | 89.08$\pm$0.14 | 86.11$\pm$0.27 | 79.39$\pm$0.41 | 54.49$\pm$2.77 |
| | **ANL-CE** | 91.66$\pm$0.04 | **90.02$\pm$0.23** | **87.28$\pm$0.02** | **81.12$\pm$0.30** | **61.27$\pm$0.55** |
| | **ANL-FL** | 91.79$\pm$0.19 | **89.95$\pm$0.20** | **87.25$\pm$0.11** | **81.67$\pm$0.19** | **61.22$\pm$0.85** |
| CIFAR-100 | CE | 71.14$\pm$0.38 | 55.97$\pm$1.11 | 40.72$\pm$0.74 | 22.98$\pm$0.07 | 7.55$\pm$0.21 |
| | FL | 71.02$\pm$0.36 | 55.94$\pm$0.53 | 39.55$\pm$1.24 | 23.21$\pm$0.49 | 7.80$\pm$0.27 |
| | MAE | 7.35$\pm$1.19 | 7.91$\pm$0.66 | 3.61$\pm$0.21 | 3.63$\pm$0.35 | 2.83$\pm$1.35 |
| | GCE | 61.62$\pm$0.43 | 61.50$\pm$1.50 | 56.46$\pm$0.95 | 46.27$\pm$1.30 | 19.51$\pm$0.86 |
| | SCE | 70.80$\pm$0.37 | 55.04$\pm$0.37 | 39.84$\pm$0.19 | 21.97$\pm$0.92 | 7.87$\pm$0.48 |
| | NLNL | 68.72$\pm$0.60 | 46.99$\pm$0.91 | 30.29$\pm$1.64 | 16.60$\pm$0.90 | 11.01$\pm$2.48 |
| | NCE+MAE | 67.52$\pm$0.21 | 52.68$\pm$0.42 | 35.71$\pm$0.47 | 19.44$\pm$0.15 | 7.08$\pm$0.07 |
| | NCE+RCE | 68.22$\pm$0.28 | 64.20$\pm$0.47 | 57.97$\pm$0.30 | 46.26$\pm$1.07 | 25.65$\pm$0.51 |
| | NFL+RCE | 68.04$\pm$0.30 | 64.33$\pm$0.23 | 58.48$\pm$0.50 | 47.20$\pm$0.58 | 26.26$\pm$0.19 |
| | NCE+AEL | 64.98$\pm$0.42 | 48.13$\pm$0.32 | 32.11$\pm$0.89 | 20.75$\pm$2.00 | 7.97$\pm$1.02 |
| | NCE+AGCE | 68.61$\pm$0.12 | 65.30$\pm$0.21 | 59.74$\pm$0.68 | 47.96$\pm$0.44 | 24.13$\pm$0.07 |
| | NCE+AUL | 69.91$\pm$0.18 | 65.26$\pm$0.17 | 56.67$\pm$0.21 | 39.98$\pm$0.42 | 13.30$\pm$0.08 |
| | **ANL-CE** | 70.68$\pm$0.23 | **66.79$\pm$0.34** | **61.80$\pm$0.50** | **51.52$\pm$0.53** | **28.07$\pm$0.28** |
| | **ANL-FL** | 70.40$\pm$0.15 | **66.54$\pm$0.29** | **61.73$\pm$0.48** | **51.32$\pm$0.34** | **27.97$\pm$0.58** |

Table 7: Test accuracies (%) of different methods on benchmark datasets with asymmetric label noise ($\eta \in \{0.1, 0.2, 0.3, 0.4\}$). The results (mean±std) are reported over 3 random runs under different random seeds $(1, 2, 3)$ and the top-2 best results are **boldfaced**.

| DATASETS | METHODS | ASYMMETRIC NOISE RATE ($\eta$) | | | |
| --- | --- | --- | --- | --- | --- |
| | | 0.1 | 0.2 | 0.3 | 0.4 |
| MNIST | CE | 97.70±0.02 | 94.02±0.18 | 88.90±0.07 | 81.79±0.34 |
| | FL | 97.62±0.05 | 94.41±0.11 | 88.82±0.35 | 81.99±0.61 |
| | MAE | 99.05±0.06 | **99.11±0.03** | 98.42±0.09 | 87.40±4.01 |
| | GCE | 98.97±0.02 | 96.59±0.07 | 88.99±0.27 | 81.91±0.58 |
| | SCE | 99.05±0.02 | 97.95±0.23 | 94.00±0.41 | 84.54±0.14 |
| | NLNL | 98.63±0.06 | 98.35±0.01 | 97.51±0.15 | 95.84±0.26 |
| | NCE+MAE | 99.12±0.09 | 98.33±0.14 | 95.16±0.21 | 85.79±0.47 |
| | NCE+RCE | **99.31±0.06** | 98.79±0.10 | 95.16±0.08 | 91.36±0.22 |
| | NFL+RCE | **99.27±0.02** | 98.79±0.15 | 96.97±0.09 | 91.36±0.40 |
| | NCE+AEL | 99.06±0.05 | 99.05±0.07 | 98.88±0.02 | 98.29±0.27 |
| | NCE+AGCE | 99.11±0.04 | 99.04±0.02 | **98.94±0.03** | **98.41±0.04** |
| | NCE+AUL | 99.11±0.05 | **99.09±0.03** | **98.98±0.06** | **98.70±0.01** |
| | **ANL-CE** | 99.10±0.05 | 99.04±0.04 | 98.91±0.07 | 98.01±0.10 |
| | **ANL-FL** | 99.00±0.05 | 99.05±0.09 | 98.93±0.02 | 98.18±0.01 |
| CIFAR-10 | CE | 86.85±0.15 | 83.00±0.33 | 78.15±0.17 | 73.69±0.20 |
| | FL | 86.32±0.28 | 83.03±0.10 | 78.53±0.16 | 73.78±0.16 |
| | MAE | 88.22±0.05 | 79.63±0.74 | 67.35±3.41 | 57.36±2.37 |
| | GCE | 88.19±0.21 | 85.55±0.24 | 79.32±0.52 | 72.83±0.17 |
| | SCE | 89.57±0.11 | 86.22±0.44 | 80.20±0.20 | 74.01±0.52 |
| | NLNL | 88.54±0.25 | 84.74±0.08 | 81.26±0.43 | 76.97±0.52 |
| | NCE+MAE | 88.22±0.25 | 86.16±0.18 | 82.98±0.15 | 75.23±0.24 |
| | NCE+RCE | 89.98±0.19 | 88.36±0.13 | 84.84±0.16 | **77.75±0.37** |
| | NFL+RCE | 90.11±0.06 | 88.26±0.27 | 84.72±0.18 | 77.29±0.30 |
| | NCE+AEL | 87.64±0.16 | 85.64±0.07 | 81.95±0.38 | 74.55±0.42 |
| | NCE+AGCE | 90.21±0.16 | 88.48±0.09 | 84.79±0.15 | **78.60±0.41** |
| | NCE+AUL | 90.12±0.06 | 88.29±0.15 | 84.84±0.06 | 76.99±0.26 |
| | **ANL-CE** | **90.90±0.13** | **89.13±0.11** | **85.52±0.24** | 77.63±0.31 |
| | **ANL-FL** | **90.81±0.20** | **89.09±0.31** | **85.81±0.23** | 77.73±0.31 |
| CIFAR-100 | CE | 65.17±0.45 | 58.25±1.00 | 50.30±0.19 | 41.53±0.34 |
| | FL | 64.55±0.39 | 58.00±1.38 | 50.77±0.41 | 41.88±0.57 |
| | MAE | 6.63±1.60 | 6.19±0.42 | 5.82±0.96 | 3.96±0.35 |
| | GCE | 64.29±0.90 | 59.06±0.46 | 53.88±0.96 | 41.51±0.52 |
| | SCE | 64.64±0.57 | 57.78±0.83 | 50.15±0.12 | 41.33±0.86 |
| | NLNL | 59.55±1.22 | 50.19±0.56 | 42.81±1.13 | 35.10±0.20 |
| | NCE+MAE | 60.52±0.36 | 52.92±0.50 | 44.41±0.22 | 36.71±0.16 |
| | NCE+RCE | 66.18±0.23 | 62.77±0.53 | 55.62±0.56 | 42.46±0.42 |
| | NFL+RCE | 66.16±0.44 | 63.43±0.71 | 55.63±0.37 | 42.54±0.52 |
| | NCE+AEL | 57.47±0.47 | 50.49±0.12 | 42.46±0.51 | 35.04±0.29 |
| | NCE+AGCE | 66.86±0.23 | 64.05±0.25 | 56.36±0.59 | 44.90±0.62 |
| | NCE+AUL | 66.23±0.21 | 57.79±0.40 | 47.64±0.24 | 38.65±0.30 |
| | **ANL-CE** | **68.78±0.11** | **66.27±0.19** | **59.76±0.34** | **45.41±0.68** |
| | **ANL-FL** | **68.36±0.15** | **66.26±0.44** | **59.68±0.86** | **46.65±0.04** |

### D.5 Comparisons on more real-world datasets.

**CIFAR-10N/-100N** [20] are CIFAR-10/-100 equipped with human-annotated real-world noisy labels. CIFAR-10N contains five noisy label sets with noise rates of 9.03% (Aggregate), 17.23% (Random 1), 18.12% (Random 2), 17.64% (Random 3) and 40.21% (Worst). CIFAR-100N contains one noisy label set with noise rate of 40.20% (Noisy).

We use the same experimental settings and parameters as CIFAR-10 and CIFAR-100 for CIFAR-10N and CIFAR-100N, since the only difference is the noise label distribution. The results are reported in Table 8 and Table 9.

**Animal-10N** [21] is a real-world noisy data set of human-labeled online images for 10 confusing animals, with $50,000$ training and $5,000$ testing images, and its noise rate was estimated at 8%.

We follow the experimental setting in previous works [21]. We use VGG-19 with batch normalization. The SGD optimizer is employed. We train the network for 100 epochs and use an initial learning rate of 0.1, which is divided by 5 at 50% and 75% of the total number of epochs. Batch size is set to 128. Typical data augmentations including random horizontal flip are applied.

We compare our ANL-CE with CE and GCE. We use L2 regularization (weight decay) for GCE, and L1 regularization for ANL-CE. We denote the regularization coefficient by $\delta$. We tune the parameters $\{\delta\}$, $\{q, \delta\}$, $\{\alpha, \beta, \delta\}$ for CE, GCE and ANL-CE respectively. We use the best parameters $\{1 \times 10^{-3}\}$, $\{0.5, 1 \times 10^{-4}\}$, $\{0.5, 1.0, 1 \times 10^{-6}\}$ for each method in our experiments. The results are reported in Table 10.

Moreover, we experiment with NCE+RCE on this dataset and tune the parameters $\{\alpha, \beta, \delta\}$, but we find that the performance is very poor for some unknown reason. The best test accuracy we achieve is 28.28% with $\{10.0, 0.1, 5 \times 10^{-6}\}$. Since this result is too low and inconsistent with its performance on other datasets, we do not include it in the table for comparison.

**Clothing-1M** [22] is a large-scale clothing dataset contains 14 categories and 1 million training samples with nearly 40% mislabeled samples.

We follow the experimental setting in previous works [31]. We use the $14k$ and $10k$ clean data for validation and test, respectively, and we do not use the $50k$ clean training data. We use ResNet-50 pre-trained on ImageNet. For preprocessing, we resized the images to $256 \times 256$, performed mean subtraction, and cropped the middle $224 \times 224$. We use SGD with a momentum of 0.9, a weight decay of $1 \times 10^{-3}$, and batch size of 32. We train the network for 10 epochs with learning rate $1 \times 10^{-3}$ and $1 \times 10^{-4}$ for 5 epochs each. Typical data augmentations including random horizontal flip are applied.

We compare our ANL-CE with CE, GCE and NCE+RCE. In this experiment, we use L2 regularization (weight decay) for ANL-CE and set the coefficient the same as those for CE, GCE and NCE+RCE. We tune the parameters $\{q\}$, $\alpha, \beta$ and $\{\alpha, \beta\}$ for GCE, NCE+RCE and ANL-CE respectively. We use the best parameters $\{0.6\}$, $\{10.0, 1.0\}$ and $\{5.0, 0.1\}$ for each method in our experiments. The results are reported in Table 11.

Table 8: Test accuracies (%) of different methods on CIFAR-10N dataset. The results (mean±std) are reported over 3 random runs under different random seeds $(1, 2, 3)$ and the top-2 best results are **boldfaced**.

| Methods | Clean | Aggregate | Random 1 | Random 2 | Random 3 | Worst |
|---|---|---|---|---|---|---|
| CE | 90.38±0.11 | 85.09±0.30 | 79.09±0.28 | 78.59±0.42 | 78.39±0.50 | 61.43±0.52 |
| GCE | 89.66±0.20 | 87.38±0.07 | 85.87±0.27 | 85.43±0.13 | 85.51±0.15 | 75.19±0.23 |
| SCE | 91.38±0.12 | 88.48±0.26 | 85.65±0.30 | 85.71±0.19 | 85.87±0.13 | 73.65±0.29 |
| NCE+RCE | 90.94±0.01 | 89.17±0.28 | 87.62±0.34 | **87.66±0.12** | **87.70±0.18** | 79.74±0.09 |
| NCE+AGCE | 91.08±0.06 | **89.27±0.28** | **87.92±0.02** | 87.61±0.20 | 87.62±0.16 | **79.91±0.37** |
| ANL-CE (ours) | 91.66±0.04 | **89.66±0.12** | **88.68±0.13** | **88.19±0.08** | **88.24±0.15** | **80.23±0.28** |

Table 9: Test accuracies (%) of different methods on CIFAR-100N dataset. The results (mean±std) are reported over 3 random runs under different random seeds $(1, 2, 3)$ and the top-2 best results are **boldfaced**.

| Methods | Clean | Noisy |
|---|---|---|
| CE | 71.14±0.38 | 48.63±0.53 |
| GCE | 61.62±0.43 | 50.97±0.60 |
| SCE | 70.80±0.37 | 48.52±0.11 |
| NCE+RCE | 68.22±0.28 | 54.27±0.09 |
| NCE+AGCE | 68.61±0.12 | **55.96±0.20** |
| ANL-CE (ours) | 70.68±0.23 | **56.37±0.42** |

Table 10: Test accuracies (%) of different methods on Animal-10N dataset. The results (mean±std) are reported over 3 random runs under different random seeds $(1, 2, 3)$ and the top-1 best results are **boldfaced**.

| Methods | CE | GCE | ANL-CE (ours) |
|---|---|---|---|
| Test Acc. (%) | 78.92±0.76 | 80.39±0.17 | **80.72±0.37** |

Table 11: Test accuracies (%) of different methods on Clothing-1M dataset. The top-1 best results are **boldfaced**.

| Methods | CE | GCE | NCE+RCE | ANL-CE (ours) |
|---|---|---|---|---|
| Test Acc. (%) | 68.07 | 68.94 | 69.07 | **69.93** |

