# OpenReview forum: "Active Negative Loss Functions for Learning with Noisy Labels"
_NeurIPS.cc/2023/Conference — NeurIPS 2023 poster_

### Official Review · Reviewer_abGG · 2023-06-23

**Soundness:** 3 good
**Presentation:** 3 good
**Contribution:** 3 good
**Rating:** 7
**Confidence:** 3

**Summary:**

In this work, the authors propose a new class of theoretically robust passive loss functions named Normalized Negative Loss Functions (NNLFs). By replacing the MAE in APL with the proposed NNLFs, this paper improves APL and proposes a new framework called Active Negative Loss (ANL).

**Strengths:**

1. The motivation is clearly stated. The writing of this paper is also clear.
2. The theoretical study in this paper is solid.

**Weaknesses:**

1. More real-world datasets are needed for algorithm verification.
2. This paper contains many typos.

**Questions:**

1. Some mathematical notations should be clearly defined, such as N in Section 2.1.
2. The authors should carefully proofread their paper. Some typos should be corrected, such as "at at least one other class position...". In supplementary material, the equation index is missing in "So, from Eq () we have..." above Eq. 25.
3. This paper only investigates the proposed method on one real-world dataset, namely WebVision, which I think is not adequate. In fact, there are many benchmark datasets with real-world noise, such as Animal-10N, Clothing-1M, etc.

**Limitations:**

See my comments above.

---

> ### Author Rebuttal · Authors · 2023-08-09
>
> We appreciate the suggestion from the reviewer and will incorporate it into the updated version of our paper.
>
> 1. Typos and mathematical notations.
>
>    We thank the reviewer for pointing out the typos and the unclear mathematical notations. We will check our paper carefully and correct the relevant errors. We apologize for the inconvenience caused by this.
>
> 2. More experiments.
>
>    Please see the global response.

---

> > ### Comment · Reviewer_abGG · 2023-08-12
> > **I have read the authors' rebuttal**
> >
> > Thanks for the authors' rebuttal. I do not have major concerns on this paper, and vote for an acceptance.

---

### Official Review · Reviewer_btFc · 2023-07-06

**Soundness:** 3 good
**Presentation:** 3 good
**Contribution:** 3 good
**Rating:** 7
**Confidence:** 4

**Summary:**

This paper proposes a robust loss function for learning from label noise. The authors first find that negative losses proposed in APL are scaled versions of MAE, which is not training-friendly. To solve this problem, the authors propose the Active Negative Loss (ANL) framework which improves the APL loss by replacing the passive losses with a newly designed set of negative losses. Theoretically, the ANL losses are proved to be noise tolerant and the designed negative loss is shown to focus more on the well-learned classes and samples compared with MAE. Experimentally, the ANL framework outperforms the state-of-the-art robust losses on multiple datasets.

**Strengths:**

- This paper is well-written and easy to read.
- This paper provides new insights into the negative losses proposed in APL.
- The proposed loss framework is provably robust and its gradient is theoretically understood.
- The experiments are extensive and sound.

**Weaknesses:**

- In Section 3.1, the motivation for incorporating the three components in NNLF is lacking. Why do we need the three components in NNLF? And why do the three components make NNLF a good robust loss function?
- The theoretical analysis is less original. Most discussions of the relationship between symmetric loss function and noise tolerance can be found in previous works, e.g. APL. Accordingly, the proofs of Theorems 2, 3, and 4 could be simplified.
- Typos. Lines 196, 201, 220. "theorem" should start with a capital T.

**Questions:**

- What motivates the three components of NNLF?
- In Eq. (7), is $A$ a constant over all $p(k|x)$? If not, the derivation of gradients in Eq. (15) should consider the dependency of $A$ on $p(k|x)$.
- The proposed ANL framework performs less well on the MNIST dataset (Tables 2, 6, 7). Are there any explanations?

---

> ### Author Rebuttal · Authors · 2023-08-09
>
> We appreciate the suggestion from the reviewer and will incorporate it into the updated version of our paper.
>
> 1. The three components of NNLF.
>
>    Our goal is to create a new loss function to replace the MAE in APL. This means that: 1. The loss function must conform to the definition of passive loss functions in APL. 2. The loss function is different from MAE and does not treat all samples equally, which leads to training difficulties. 3. The loss must be robust to noisy labels. The three components of NNLF: complementary label learning, “vertical flipping”, and normalization operation correspond to these three motivations respectively. We believe that NNLF is better than MAE because it can make the model pay more attention to clean samples that have already been memorized, as described in our discussion of Theorem 5 and Theorem 6. Theorem 5 and Theorem 6 are due to the “vertical flipping” operation, which means that the gradient increases as the prediction probability of non-labeled classes decreases.
>
> 2. The theoretical analysis of noise tolerance is less original.
>
>    We appreciate your comment on the originality of our discussion on noise tolerance. We kept this part for the sake of completeness of our proofs, but we agree that it can be simplified. We will revise our paper accordingly.
>
> 3. Is $A$ a constant ?
>
>    Yes, $A$ is a constant.
>
> 4. ANL framework performs less well on the MNIST.
>
>    We appreciate the reviewer’s insightful question. We conduct experiments on the MNIST dataset with 0.8 symmetric noise rate using ANL-CE and NCE+AGCE, and calculate the entropy $H=-\sum_i p(i|x) \log p(i|x)$ of the model’s prediction probability on clean samples at each step. The lower the entropy, the closer the prediction probability is to a one-hot vector, indicating that the model is more confident in its prediction results. We find that in the first 5 epochs, the entropy values of both methods are very close and have similar trends. After 5 epochs, the entropy of NCE+AGCE stabilizes at around 0.09 and keeps oscillating, while that of ANL-CE shows a continuous decreasing trend and approaches 0. This indicates that our model is too confident in its prediction results.
>
>    We believe that this phenomenon is caused by our Theorem 5 and Theorem 6, which force the loss function to continuously make the model learn already learned samples, causing overfitting of the model to clean samples in the training set and loss of generalization performance. Although we use L1 regularization to alleviate this problem, considering that MNIST is a very simple dataset, the effectiveness of L1 regularization may be affected. As described in the Limitations section, we believe that regularization methods are the main limitation of our method.
>
> 5. Typos
>
>    Thank you for pointing out the typos in our paper. We will carefully check our paper and correct all the spelling and grammar errors. We apologize for the inconvenience caused by this.

---

> > ### Comment · Reviewer_btFc · 2023-08-18
> >
> > Thanks for the rebuttal. The reply solved my concerns. I will keep my score and vote for acceptance.

---

### Official Review · Reviewer_u2rQ · 2023-07-07

**Soundness:** 3 good
**Presentation:** 3 good
**Contribution:** 3 good
**Rating:** 6
**Confidence:** 5

**Summary:**

This paper proposes robust loss function to improve training of DNNs using noisy labels. Previously proposed work of Active Passive Loss functions is improved.

**Strengths:**

Paper is well written and contains theoretical foundation of the proposed work. There are a lot of derivations which give a broad understanding of loss functions. Table 2 shows improved performance of the proposed loss functions compared to [6] and [18] in some cases. Though the improvement is not consistent.

**Weaknesses:**

1. The  terminology used in this paper is mainly based on previous work of reference [6]. The terms like active and passive loss functions do not clearly separate the loss functions. As shown in the appendix, MAE=1(1-p(y|x)), shows that MAE also aims to maximise p(y|x) therefore it is also an active loss function. In [6], MAE was considered as a passive loss function which is not correct Mathematically in cases when we are dealing with a prediction as \sum_k{p(k|x)}=1. This paper also inherits the same definitions from [6].

2. Changing the sign of a loss function also changes the max to min. If we are maximising L, we should be minimising -L. Therefore change of sign may not be considered making any fundamental difference. The term `Vertical Flip' is. derived from the graph. Mathematically, there is no thing like a vertical flip.

3. Definition of A in Eq (7) is not clear. How a loss can be computed between a vector [ . .........p_min, .........] and y ? In the appendix, A is defined as a constant value.

4. As shown in the appendix, the proposed normalisation  by [6] is  scaling and the same is inherited in the current work. If all loss values are scaled by the same number then how it will be robust to noise? The loss generated by the noisy and the clean samples will be scaled equally and noisy labels are not known at the time of training.

5. In Eq 9, denominator can be simplified as: \sum_k{A+ log p(k|x)}=KA+1, reducing Eq 9 to: NNFL=1-A/(KA+1)-logp(y|x)/(1+KA) . It is again a scaled version of the original function added a constant value. What fundamental change makes it robust to noise?

6. The range of asymmetric noise is kept quite small {.2,.3.4} compared to symmetric noise {.4,.6,.8} in Table 2. A larger range would have revealed more insights.


**Questions:**

The same as above.

**Limitations:**

More theoretical insights are required for the proposed loss functions.

---

> ### Author Rebuttal · Authors · 2023-08-09
>
> We appreciate the suggestion from the reviewer and will incorporate it into the updated version of our paper.
>
> 1. Active and passive loss functions.
>
>    We agree that there is no clear distinction between active and passive loss functions in [1]. Since our work follows [1], we directly use these definitions. Regarding why to distinguish between active and passive loss functions, we suggest referring directly to [1], which has a detailed discussion on this.
>
> 2. Vertical flip.
>
>    Firstly, the name “vertical flip” does come directly from the graph because it is very intuitive. Secondly, taking CE as an example, $A-(-\log p(i|x))$ obtained by “vertical flip” is not the same as $-(-\log p(i|x))$ obtained by simply changing the sign because we do not directly minimize it but also apply normalization to it. Specifically, after normalizing $\sum_{i \ne y}^K-(-\log p(i|x))$, we get $\frac{1}{K-1}\cdot(1-\frac{-\log p(y|x)}{\sum_k - \log p(k|x)})$, which shows that when we minimize it, we are actually minimizing $p(y|x)$ and maximizing $p(i\ne y|x)$, which contradicts our optimization goal. In contrast, after normalizing $\sum_{i \ne y}^K A - (- \log p(i|x))$, we get $\frac{1}{K-1}\cdot (1-\frac{A-(-\log p(y|x))}{\sum_k A - (-\log p(k|x))})$, which shows that when we minimize it, we are actually maximizing $p(y|x)$ and minimizing $p(i\ne y|x)$, which is consistent with our optimization goal.
>
> 3. Definition of $A$.
>
>    Thanks to the reviewer for pointing out that the definition of $A$ is unclear. $A$ is indeed a constant, and we will revise its definition to make it more clear and precise.
>
> 4. Normalization.
>
>    The normalization proposed by [1] does not mean that all the loss values are scaled by the same number. For example, for $NCE=\frac{1}{\sum_k - \log p(k|x)} \cdot \big(- \log p(y|x)\big)$, the factor $\frac{1}{\sum_k - \log p(k|x)}$ varies for different samples $x$ with different prediction probabilities $p(i|x), i \in [1, \cdots, K]$. Moreover, the noise robustness of the normalized loss function is determined by its symmetry: $\sum_{y=1}^K \Big( \frac{1}{\sum_k - \log p(k|x)} \cdot \big(- \log p(y|x)\big)\Big) = 1$, and [2] has proved that the symmetric loss functions are robust to noise.
>
> 5. The denominator of Eq (9).
>
>    The denominator of Eq (9) can be expressed as: $\sum_k (A + \log p(k|x) ) = K \cdot A + \sum_k \log p(k|x)$. We know that $\sum_k p(k|x)=1$ and $p(k|x) \in [0, 1]$. Therefore, we can conclude that $\sum_k \log p(k|x) \in [-\infty, K\cdot \log \frac{1}{K}]$ and $\sum_k ( A + \log p(k|x) ) \ne K \cdot A + 1$.
>
> 6. The range of asymmetric noise.
>
>    For synthetic asymmetric noise, we often mislabel one class as a specific other class. For instance, on CIFAR-10 with synthetic asymmetric noise, we randomly label bird images as planes with probability $\eta$ (noise rate). If $\eta$ is larger than 0.5, it means that most of the birds are labeled as planes, which is very rare in the real world. To the best of our knowledge, no previous work has experimented with synthetic asymmetric noise at noise rates $\eta$ greater than 0.5.
>
> [1] Normalized Loss Functions for Deep Learning with Noisy Labels, ICML, 2020
>
> [2] Robust Loss Functions under Label Noise for Deep Neural Networks, AAAI, 2017

---

> > ### Comment · Reviewer_u2rQ · 2023-08-12
> >
> > The authors have followed the active and passive terminology used by previous papers. The intuition of why the normalisation will result in robustness will improve the paper.  The demonstration of the proposed loss is perhaps for DNNs while it remains unclear if it will be effective for other deep architectures such as Vision Transformers which use attention mechanism. Networks architectures used in different experiments must be mentioned. Nevertheless, the paper may be considered for publication.

---

### Official Review · Reviewer_j1Cc · 2023-07-08

**Soundness:** 3 good
**Presentation:** 3 good
**Contribution:** 2 fair
**Rating:** 5
**Confidence:** 4

**Summary:**

This paper introduces a novel type of loss function called Active Negative Loss (ANL), which builds upon the Active Passive loss function (APL) framework. The authors identify a limitation in APL, where the passive loss function, being a scaled version of Mean Absolute Error (MAE), can lead to slower convergence and underfitting issues. To address this, the authors propose replacing the passive loss in APL with a normalized negative loss, drawing inspiration from negative learning and vertical flipping techniques. The paper also provides theoretical justifications for the superiority of ANL over APL. The proposed ANL framework is evaluated through experiments conducted on CIFAR10 and CIFAR100 datasets with synthetic label noise, as well as ILSVRC12 and WebVision datasets with real-world label noise.

**Strengths:**

1. The motivation is clearly articulated and accompanied by well-explained reasoning.

2. The proposed ANL framework can be considered an enhanced version of the APL framework, offering improved performance with theoretical backing. Specifically, it demonstrates robustness against both symmetric and asymmetric label noise under certain conditions.

3. Code is provided to facilitate reproducibility.


**Weaknesses:**

1. Although the proposed loss is theoretically sound, many of the techniques or analyses employed are not very new. The ANL framework appears to be a simple combination of the NL loss [R1] and APL loss [R2]. While I acknowledge the validity of this combination, I do not believe it constitutes a significant contribution to the LNL community.

2. The authors mention that regularization, such as L1 regularization, is applied to the ANL framework. I am curious to know if other approaches in your experiments also utilize this regularization. Additionally, the absence of ablation studies investigating the impact of regularization on the ANL framework is worth considering.

3. I believe conducting further experiments to verify the effectiveness of the ANL framework is necessary. For instance, experiments involving CIFAR with controlled instance-dependent label noise, CIFAR-10N, CIFAR-100N, and the Clothing1m dataset would provide valuable insights.


[R1] NLNL: Negative Learning for Noisy Labels

[R2] Normalized Loss Functions for Deep Learning with Noisy Labels

**Questions:**

See **weaknesses**

**Limitations:**

I think this work does not bring negative societal impact.

---

> ### Author Rebuttal · Authors · 2023-08-09
>
> We appreciate the suggestion from the reviewer and will incorporate it into the updated version of our paper.
>
> 1. A simple combination of the NL and APL.
>
>    First of all, although our ANL may seem like a simple combination of existing techniques, we have a strong motivation for doing so: finding a passive loss function that performs better than MAE to enhance APL.
>
>    Secondly, our NLF and NL are still fundamentally different. Take NLF(CE) as an example:
>    $$
>    L_\text{NLF}=\sum_{k=1}^K (1-q(k|x))\big(A-(-\log (p(k|x))\big).
>    $$
>    And the NL loss is:
>    $$
>    L_\text{NL}=\sum_{k=1}^K (1-q(k|x)) \big( - \log (1-p(k|x)) \big).
>    $$
>    Although the $q(k|x)$ in NL is actually obtained by randomly selecting complementary labels rather than directly from the given labels as we did, we use the same form for comparison purposes.
>
>    Next, let's take a closer look at the gradient of these two losses on $p(j|x), j \ne y$. For NLF, we have:
>    $$
>    \frac{\partial}{\partial p(j|x)} L_\text{NLF} = \frac{1}{p(j|x)}.
>    $$
>    And for NL loss, we have:
>    $$
>    \frac{\partial}{\partial p(j|x)} L_\text{NL} = \frac{1}{1-p(j|x)}.
>    $$
>    We can see that for NL loss, its gradient decreases as $p(j|x)$ decreases. However, for our NLF, the gradient increases. This means that our NLF can keep focusing on well-learned samples. This property holds even after normalization (as shown in Theorem 5 and 6). In fact, if we were to use NL instead of our NLF, Theorem 5 and 6 would not hold.
>
> 2. Regularization.
>
>    For the experiments of other methods, we followed their original papers and used L2 regularization. However, to ensure fairness and investigate whether other regularizations would have positive effects on other methods, we also conducted a set of experiments using L1 regularization for APL. The details and results of these experiments are presented in Appendix on L530. We found that changing the regularization methods did not significantly affect the performance of APL. Moreover, regarding the ablation study of the impact of regularization on the ANL framework, we have already done the relevant experiments and discussions on L247 of the paper.
>
> 3. More experiments.
>
>    Please see the global response.

---

> > ### Comment · Reviewer_j1Cc · 2023-08-18
> > **Thank you for your prompt response and thorough explanation.**
> >
> > The majority of my concerns have been adequately addressed. However, I would like to note that, in my opinion, the performance exhibited in both the main paper and the Rebuttal phase does not match the competitiveness of state-of-the-art methods like the ELR loss [R1]. Nevertheless, this work does make contributions in analyzing robust losses within the domain of Learning with Noisy Labels (classification). In light of this, I decide to maintain my rating at 5. I strongly encourage the authors to include rebuttal experiments involving additional methods such as ELR, Peer Loss in the upcoming version.
> >
> > [R1] Early-Learning Regularization Prevents Memorization of Noisy Labels

---

### Author Rebuttal · Authors · 2023-08-09

Following the suggestions of the reviewers and in order to further validate the effectiveness of our method, we conducted a set of experiments on CIFAR-10N [1], CIFAR-100N [1], Animal-10N [2], and Clothing-1M [3]. For some experiments, we can only compare a few methods due to time constraint. The table of the experimental results can be found in the pdf file. Overall, our method outperforms all the other compared methods on all four datasets.

**CIFAR-10N and CIFAR-100N**

We use the same experimental settings and parameters as CIFAR-10 and CIFAR-100 for CIFAR-10N and CIFAR-100N, since the only difference is the noise label distribution. The results are reported in Table 1 and Table 2.

**Animal-10N**

We follow the experimental setting in previous works [2]. We use VGG-19 with batch normalization. The SGD optimizer is employed. We train the network for 100 epochs and use an initial learning rate of 0.1, which is divided by 5 at 50% and 75% of the total number of epochs. Batch size is set to 128. Typical data augmentations including random horizontal flip are applied.

We compare our ANL-CE with CE and GCE. We use L2 regularization (weight decay) for GCE, and L1 regularization for ANL-CE. We denote the regularization coefficient by $\delta$. We tune the parameters $\\{\delta\\}, \\{q, \delta\\}$ and $\\{\alpha, \beta, \delta\\}$ for CE, GCE and ANL-CE respectively. We use the best parameters $\\{10^{-3}\\}, \\{0.5, 10^{-4}\\}, \\{0.5, 0.1, 10^{-6}\\}$ for each method in our experiments.  The results are reported in Table 3.

Moreover, we experiment with NCE+RCE on this dataset and tune the parameters $\\{\alpha,\beta,\delta\\}$, but we find that the performance is very poor for some unknown reason. The best test accuracy we achieve is 28.28 with $\\{10.0, 0.1, 5 \times 10^{-6}\\}$. Since this result is too low and inconsistent with its performance on other datasets, we do not include it in the table for comparison.

**Clothing-1M**

We follow the experimental setting in previous works [4]. We use the $14k$ and $10k$ clean data for validation and test, respectively, and we do not use the $50k$ clean training data. We use ResNet-50 pre-trained on ImageNet. For preprocessing, we resized the images to $256 \times 256$, performed mean subtraction, and cropped the middle $224 \times 224$. We used SGD with a momentum of 0.9, a weight decay of $10^{-3}$, and batch size of 32. We train the network for 10 epochs with learning rate $10^{-3}$ and $10^{-4}$ for 5 epochs each. Typical data augmentations including random horizontal flip are applied.

We compare our ANL-CE with CE, GCE and NCE+RCE. In this experiment, we use L2 regularization (weight decay) for ANL-CE and set the coefficient the same as those for CE, GCE and NCE+RCE. We tune the parameters $\\{q\\}, \\{\alpha, \beta\\}$ and $\\{\alpha, \beta\\}$ for GCE, NCE+RCE and ANL-CE respectively. We use the best parameters $\\{0.6\\}, \\{10.0, 1.0\\}$ and $\\{5.0, 0.1\\}$ for each method in our experiments. The results are reported in Table 4.


[1] Learning with Noisy Labels Revisited: A Study Using Real-World Human Annotations, ICLR, 2022

[2] SELFIE: Refurbishing Unclean Samples for Robust Deep Learning, ICML, 2019

[3] Learning From Massive Noisy Labeled Data for Image Classification, CVPR, 2015

[4] Joint Optimization Framework for Learning with Noisy Labels, CVPR, 2018

---

### Decision · Program_Chairs · 2023-09-21

**Decision:**

Accept (poster)

**Comment:**

This paper presents the Active Negative Loss (ANL) framework, which aims to improve the robustness of deep neural networks against noisy labels by introducing a new class of passive loss functions called Normalized Negative Loss Functions (NNLFs). The paper has several strengths, such as a clear motivation, having theoretical foundations, and code availability. The theoretical foundations seem solid and the empirical results are promising. However, there also remain some notable weaknesses. All the reviewers unanimously agree that the paper bears a strong technical contribution. Thus, the AC decides to accept this paper.